# One Stone Three Birds: Training-free Core-context-aware Attention for Efficient LLM Prefilling, Decoding, and KV Caching

## Abstract

The quadratic computational complexity of self-attention poses a critical bottleneck for large language models (LLMs) processing ultra-long contexts. While various sparse attention and KV cache compression methods have been proposed to improve efficiency, they often suffer from limitations such as reliance on fixed patterns, inability to handle both prefilling and decoding stages, or the requirement for additional training. In this paper, we propose Training-free Core-context-aware Attention (TFCA-Attention), a training-free sparse attention mechanism that achieves "one stone three birds": it unifies acceleration for prefilling, decoding, and KV cache reduction through a consistent sparsity mechanism. TFCA-Attention features an offline calibration phase that determines head-specific sparsity budgets and an online token selection phase that adaptively retains core context tokens using a lightweight redundancy metric. Theoretically, we provide a bounded approximation error guarantee, ensuring long context modeling accuracy. Extensive experiments demonstrate that TFCA-Attention achieves a **2.8×** speedup and reduces KV cache by **61%** at 128K context length while maintaining performance comparable to full attention across various benchmarks, offering a practical plug-and-play solution for efficient long-context inference.

## 1 Introduction

Large language models (LLMs) (Touvron et al., 2023a;b; Brown et al., 2020a; Wei et al., 2022) like GPT-o1 (OpenAI, 2024) and DeepSeek-R1 (Guo et al., 2025) have become the cornerstone of modern natural language processing, demonstrating remarkable capabilities in tasks requiring long-context understanding, such as multi-step reasoning (Brown et al., 2020b; Singhal et al., 2025) and document-level comprehension (Cao et al., 2017; Pasunuru et al., 2021). This success is largely attributed to the self-attention mechanism (Vaswani et al., 2017), which enables modeling dependencies across entire sequences. However, as context lengths extend to extremes (*e.g.*, 128K), the quadratic complexity of self-attention becomes a critical bottleneck. Moreover, the presence of redundant tokens, which contribute minimally to the final output, dilutes attention to critical information, degrading efficiency and accuracy (Jiang et al., 2024; Chen et al., 2025).

To address these challenges, researchers have pursued several distinct approaches. Early efforts, such as *static sparse attention* Zaheer et al. (2020); Beltagy et al. (2020); Xiao et al. (2024), employ predefined attention patterns to reduce computation, yet they lack adaptability to input content and often compromise performance on context-sensitive tasks. More recent *prefilling-stage dynamic methods* Jiang et al. (2024); Lai et al. (2025); Xu et al. (2025) adjust attention patterns per head during prefilling, but they are restricted to a limited set of handcrafted sparsity patterns and do not address inefficiency in the decoding phase. On the other hand, *decoding-stage KV cache compression techniques* Hao et al. (2025); Li et al. (2024); Qin et al. (2025); Behnam et al. (2025) reduce memory usage by evicting or merging cached KV entries, yet they fail to accelerate prefilling and typically apply uniform compression strategies across attention heads, ignoring the intrinsic diversity of redundancy across heads. While a few unified frameworks (Xiao et al., 2025; Yang et al., 2025; Gao et al., 2024; 2025) target both prefilling and decoding, they require continued training to determine head-specific patterns or learn block-sparse patterns.

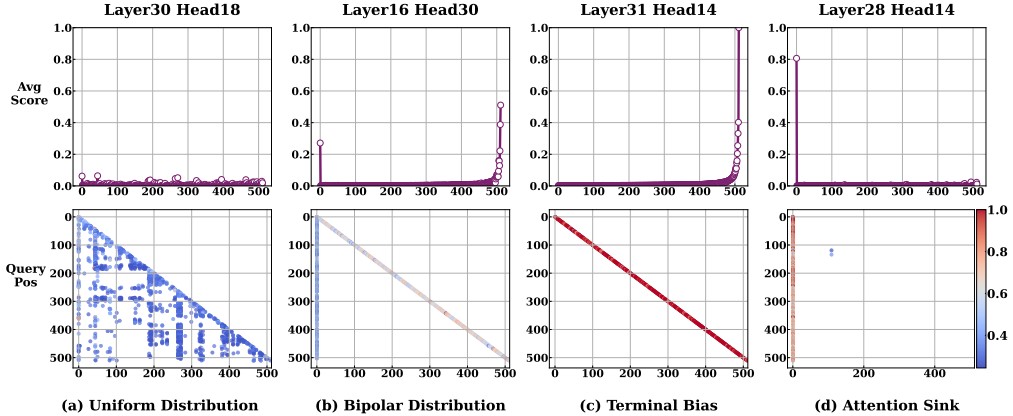

Figure 1: Visualizations of attention distributions in LLaMA-3.1-8B-Instruct: 1) average attention scores across key positions on the first row, and 2) scatter points highlighting attention scores above row-wise averages on the second row. The observations motivate the design principles of TFCA-Attention (see Appendix B for more visualizations).

The fundamental limitations of existing work can be summarized as two critical gaps: First, no existing method provides a comprehensive *training-free* solution that dynamically accelerates both prefilling and decoding while simultaneously reducing KV cache footprint without requiring architectural modifications or parameter updates. Second, they fail to model the intricate nature of attention redundancy adequately. As demonstrated in Figure 1, different attention heads exhibit diverse redundancy distributions across layers and models, ranging from uniform distributions to attention sinks, aligning with findings in prior works (Beltagy et al., 2020; Xiao et al., 2024; Jiang et al., 2024). Meanwhile, token importance varies dynamically within input content. However, existing methods either use uniform compression strategies or rely on a limited set of hand-designed patterns, lacking the fine-grained, input-adaptive mechanism required to handle such head-specifc and context-dependent redundancy.

To address these limitations, we propose Training-free Core-context-aware Attention (TFCA-Attention), a training-free dynamic attention mechanism that achieves "one stone three birds" by accelerating *prefilling*, *decoding*, and reducing *KV cache* simultaneously. In contrast to prior unified approaches (Xiao et al., 2025; Yang et al., 2025; Gao et al., 2024; 2025), TFCA-Attention requires no parameter updates, architectural adjustments, or training data. It operates in two phases: 1) An offline sparsity configuration determination phase determines a head-specific sparsity budget by estimating the redundancy level for each head. 2) An online core context selection phase dynamically selects a subset of core tokens per head based on a lightweight redundancy metric. This dual-phase design ensures our method is both head-aware and context-aware, enabling fine-grained, input-dependent acceleration without any finetuning or retraining. Furthermore, we provide a theoretical guarantee for TFCA-Attention, proving that its approximation error is bounded and can be explicitly controlled, thus ensuring reliability. Our contributions are summarized as:

- We propose TFCA-Attention, a training-free sparse attention mechanism that unifies acceleration for prefilling, decoding, and KV cache reduction via adaptive token allocation. Unlike prior unified approaches, our method eliminates any training overhead and dynamically adapts sparsity to input content, enabling plug-and-play deployment for efficient long-context inference.

- Our proposed TFCA-Attention integrates: (1) an offline calibration phase for head-specific sparsity configuration, and (2) an online token selection phase with a lightweight, context-aware redundancy metric. This design makes TFCA-Attention dynamically adapt to the two fundamental, yet underexploited, properties of redundancy (head-specific and context-dependent redundancy).

- We provide theoretical error bounds for our approximation, and extensive experiments demonstrate that our method achieves a **2.8×** speedup and reduces KV cache by **61%** at 128K context length while maintaining performance comparable to full attention.

## 2 RELATED WORKS

**Long-Context Language Models**. Handling long input sequences is critical for applications requiring document-level reasoning, code generation, or extended dialogue history(Liu et al., 2024;

Bai et al., 2023; Hsieh et al., 2024). Some methods focus on data-driven adaptation, where models are retrained or fine-tuned on ultra-long context datasets to extend their effective context length (Fu et al., 2024; Xiong et al., 2024). These methods are computationally expensive and data-dependent. Others modify positional encoding mechanisms, such as Rotary Position Embeddings (RoPE) (Su et al., 2024), via interpolation (Chen et al., 2023), dynamic scaling (Peng & Quesnelle, 2023), or learned biases (Peng et al., 2024). While these enhance context extrapolation, they also require retraining. Alternative strategies (Xu et al., 2024; Tworkowski et al., 2023) employ memory or retrieval mechanisms to reduce computation, but often at the cost of losing fine-grained local context.

**Efficient Attention**. Existing efficient attention methods fall into two categories: *sparse attention* and *KV cache compression*. *Sparse attention* methods reduce the quadratic cost of self-attention by sparsifying the attention matrix. They either use fixed patterns across all heads (Zaheer et al., 2020; Beltagy et al., 2020; Xiao et al., 2024) or adapt patterns per head in a small set of predefined sparse patterns(Jiang et al., 2024; Lai et al., 2025; Xu et al., 2025). Crucially, these methods only accelerate the *prefilling* stage and neglect decoding. *KV cache compression* methods reduce memory during decoding by evicting or merging cached entries (Li et al., 2024; Qin et al., 2025; Wan et al., 2025; Hao et al., 2025; Behnam et al., 2025), but cannot accelerate prefilling and often apply uniform compression, ignoring head-specific redundancy. Recent studies (Chen et al., 2025; Yuan et al., 2025; Lu et al., 2025; Zhang et al., 2025; Gao et al., 2024; 2025) attempt to bridge both stages using consistent sparsity patterns. However, they often require additional model training, complex profiling, or intrusive system-level changes. Our TFCA-Attention provides a training-free, head-aware, and context-adaptive sparse attention mechanism that simultaneously accelerates prefilling and decoding.

## 3 MOTIVATIONS

### 3.1 UNDERSTANDING BOTTLENECKS OF SELF-ATTENTION IN LONG-CONTEXT MODELING

Most existing LLMs are built on the Transformer (Vaswani et al., 2017) architecture, where the self-attention mechanism serves as the core module for capturing global contextual dependencies. Given an input sequence $\mathbf{X} = \{\mathbf{x}_1, \mathbf{x}_2, \ldots, \mathbf{x}_L\} \in \mathbb{R}^{L \times d}$ of $L$ tokens with model dimension $d$, The multi-head self-attention mechanism computes contextualized representations through $h$ heads:

$$\text{MultiHead}(\mathbf{X}) = [\mathbf{Att}^1, \ldots, \mathbf{Att}^h] \in \mathbb{R}^{L \times d},$$

$$\mathbf{Att}^i = \text{softmax}\left(\frac{\mathbf{Q}^i \mathbf{K}^{i\top}}{\sqrt{d_h}}\right) \mathbf{V}_i, \tag{1}$$

$$\mathbf{Q}^i = \mathbf{X}\mathbf{W}^{Q_i}, \ \mathbf{K}^i = \mathbf{X}\mathbf{W}^{K_i}, \ \mathbf{V}_i = \mathbf{X}\mathbf{W}^{V_i},$$

where $d_h$ is the head dimension (typically $d_h = d/h$), $\mathbf{W}^{Q_i}, \mathbf{W}^{K_i}, \mathbf{W}^{V_i} \in \mathbb{R}^{d \times d_h}$ are learnable parameters for the $i$-th head. The multi-head mechanism enables parallel attention operations across distinct feature subspaces, facilitating position-aware information aggregation.

**Challenges of Self-Attention in Handling Context Redundancy**: As the context length $L$ grows, the context inevitably exhibits redundant information (Jiang et al., 2024; Chen et al., 2025; Zhang et al., 2025). Vanilla self-attention faces three challenges in handling such redundancy: 1) **Quadratic computational complexity**: It incurs $O(L^2)$ computational cost by computing pairwise attention scores across all tokens. This leads to excessive computation when much of the context is redundant. 2) **Memory growth in key-value (KV) Cache**: The KV cache, which grows linearly with $L$, presents a major deployment bottleneck. For instance, processing a 128K sequence with LLaMA2-7B requires 64GB GPU memory for KV cache, exceeding the capacity of most GPUs. 3) **Interference from irrelevant tokens**: More critically, irrelevant tokens degrade the model's ability to focus on critical information, thereby harming performance.

### 3.2 EXPLORATION AND EXPLOITATION OF ATTENTION REDUNDANCY PROPERTIES

The severe inefficiency of self-attention in long contexts raises a critical question: *given the well-observed yet underutilized properties of attention redundancy, how can we construct a unified framework that fully exploits them to accelerate both prefilling and decoding?* As established in Section 1

and visualized in Figure 1, the existence of head-specific redundancy (heterogeneous sparsity patterns across heads and layers) and context-dependent redundancy (dynamic token importance) is widely observed in prior works (Jiang et al., 2024; Xu et al., 2025; Lai et al., 2025; Xiao et al., 2024). The fundamental challenge, however, lies not in observing these properties but in addressing them simultaneously and effectively within a single, training-free acceleration framework.

Existing approaches, as categorized earlier, only address a subset of this challenge. *Static sparse methods* (Beltagy et al., 2020; Zaheer et al., 2020; Xiao et al., 2024) ignore both dynamic context and head specificity. *Prefilling-only dynamic attention* methods (Jiang et al., 2024; Lai et al., 2025) adapt per head but are constrained to a small set of hand-designed patterns and, critically, fail to accelerate decoding. *Decoding-only KV compression* methods (Hao et al., 2025; Li et al., 2024; Qin et al., 2025; Wan et al., 2025) reduce memory but cannot accelerate prefilling and often apply uniform policies across heads within a layer. While recent unified methods (Xiao et al., 2025; Yang et al., 2025; Gao et al., 2024; 2025) target both stages, they require continuous training, which hinders practical adaptation and deployment.

**Design Principles for Efficient Attention**. This fragmented landscape reveals a clear design gap: no existing method fulfills all the requirements for a truly efficient and practical long-context attention mechanism. We distill these requirements into four key design principles:

1. **Sparse Computation**: Selectively attend to a small subset of critical tokens and discard irrelevant ones due to the tokens' redundancy in both prefilling and decoding stages.

2. **Dynamic Adaptation**: The selection of critical tokens must be input-dependent, not predetermined by fixed sparse patterns.

3. **Head-aware Sparsity Customization**: Sparsity strategies must be tailored to the redundancy level of each attention head.

4. **Training-free Deployment**: Eliminates the need for parameter updates, architectural modifications, or retraining, enabling immediate plug-and-play deployment without compromising model performance.

Existing methods fulfill only a subset of these principles, creating a performance-efficiency gap. Static methods violate (2) and (3); prefill-only and decoding-only methods violate (1); existing unified approaches violate (2) and (4). This motivates our design of a unified attention mechanism that satisfies all four principles simultaneously, *i.e.*, sparsity, dynamic, head-aware, and training-free.

## 4 TRAINING-FREE CORE-CONTEXT-AWARE ATTENTION

In this paper, we propose Training-free Core-context-aware Attention (TFCA-Attention) that simultaneously accelerates prefilling, reduces decoding latency, and compresses KV cache by dynamically selecting core tokens in a head-specific and context-aware manner. Our method is completely training-free, which ensures seamless integration into existing LLMs.

**Overview of Training-free Core-context-aware Attention.** Our TFCA-Attention includes two phases: 1) During the offline sparsity configuration determination phase (Section 4.1), we adopt a small calibration dataset to estimate the redundancy level of each head and determine its appropriate sparsity configuration (i.e., the number of tokens to preserve). This phase is performed only once per model. 2) During inference, we dynamically select active tokens in each head based on the determined sparsity configuration (Section 4.2). Theoretically, we prove that the approximation error of TFCA-Attention is bounded and controllable (Appendix A). We implement our method using Triton (Tillet et al., 2019) to enable efficient parallelization across attention heads (Appendix.D.3). We illustrate the overview in Figure 2 and present the pseudo-code in Algorithms 1 and 2.

To capture both long-range and short-range dependencies, our dynamic token selection operates on two complementary components: *a global subset* and *a local subset*. We dynamically select the global subset $\mathbf{K}^G = \mathbf{K}_S$ and $\mathbf{V}^G = \mathbf{V}_S$ from the entire context based on the head-specific sparsity configuration, where $S \subseteq \{1, \ldots, L\}$ is the selected token index set. We detail the computation of $S$ in Section 4.2. This subset is responsible for modeling long-distance dependencies. On the other hand, we always preserve the most recent $w$ tokens as the local subset $\mathbf{K}^L$ and $\mathbf{V}^L$ to capture fine-grained local context, a critical element highlighted by prior works (Manakul & Gales, 2021; Yang et al., 2021; Xiao et al., 2024) and our analysis. Notably, we ensure no overlap between the

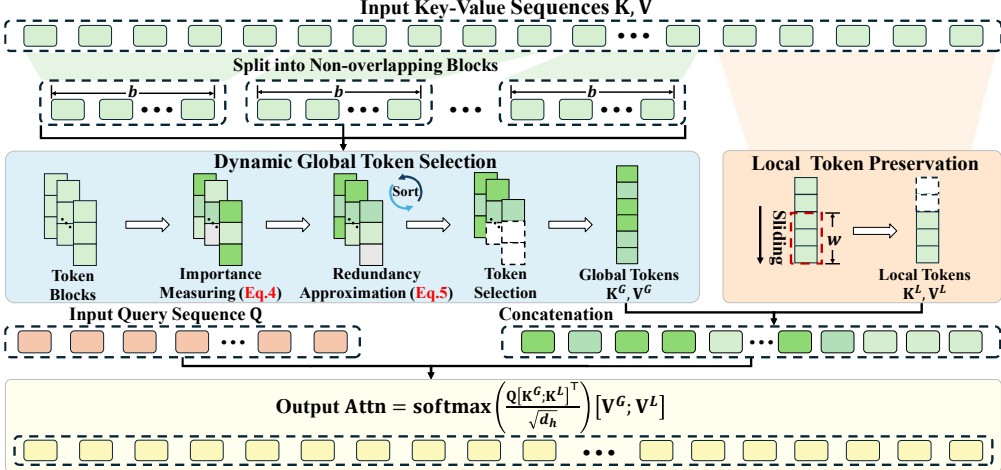

Figure 2: Overview of our TFCA-Attention. We dynamically select a subset of key/value tokens, which combines (1) a global subset $\mathbf{K}^G, \mathbf{V}^G$, selected online based on a pre-determined configuration (Sec. 4.2), to modeling long-distance dependencies; and (2) a local subset $\mathbf{K}^L, \mathbf{V}^L$, preserving neighboring tokens to capture fine-grained local context. The concatenation of these complementary subsets is used for the final attention computation in Eq. (2).

global and local subsets to avoid duplicated computation. Given the query matrix $\mathbf{Q} \in \mathbb{R}^{L \times d_h}$, our TFCA-Attention computes the attention output as follows:

$$\mathbf{Att} = \text{Softmax} \left( \frac{\mathbf{Q}[\mathbf{K}^{\mathrm{G}}; \mathbf{K}^{\mathrm{L}}]^\top}{\sqrt{d_h}} \right) [\mathbf{V}^{\mathrm{G}}; \mathbf{V}^{\mathrm{L}}]. \tag{2}$$

Since the resulting output $\mathbf{Att} \in \mathbb{R}^{L \times d_h}$ preserves the original sequence length $L$, our method seamlessly integrates into existing LLM architectures without requiring any structural modifications. For clarity and brevity, we describe all operations with respect to a single attention head in the following. We apply the same mechanism independently and in parallel to all heads in all layers.

## 4.1 OFFLINE HEAD-SPECIFIC SPARSITY DETERMINATION

Motivated by the head-specific redundancy property observed in Figure 1 and prior works (Zaheer et al., 2020; Xiao et al., 2024; Jiang et al., 2024), we introduce a one-time offline calibration to determine a unique sparsity configuration for each attention head. This configuration dictates how many tokens (token budget) the head should preserve during inference based on its inherent redundancy level. To handle a sequence $\mathbf{X}$ with the arbitrary length $L$, we operate on a block-wise basis. We partition the input sequence $\mathbf{X}$ into $m = \lfloor L/b \rfloor$ non-overlapping blocks of size $b$. Let $\mathcal{K} = \{1, 2, 4, \ldots, b\}$ denote the discrete set of allowable per-block retain counts. For each head, we seek a configuration $\mathbf{p} = [p_k | k \in \mathcal{K}]$, where $\sum_{k \in \mathcal{K}} p_k = 1$. Each $p_k \in \mathbf{p}$ represents the proportion of blocks assigned a budget of $k$ tokens, enabling an adaptive policy: the actual number of tokens preserved per block can vary based on its content, while the overall distribution is governed by $\mathbf{p}$.

**Gaussian-Sampling Configuration Candidates Generation**. Given the diverse redundancy levels across heads, we need a set of candidate configurations that smoothly transition from high-efficiency to high-accuracy modes. Creating such a diverse configuration set manually is infeasible, as it would require tuning dozens of hyperparameters across all heads. To solve this, we propose a *log-Gaussian sampling strategy* that generates candidate configurations $\mathcal{C} = \{\mathbf{p}^1, \mathbf{p}^2, \ldots, \mathbf{p}^M\}$ from efficiency-focused to accuracy-focused, controlled by just two intuitive hyperparameters. Specifically, We model the probability of retaining $k$ tokens using a log-Gaussian distribution centered around $\mu$:

$$p_k = \frac{\Phi[\log_2(k)]}{\sum_{k' \in \mathcal{K}} \Phi[\log_2(k')]}, \text{ where } \Phi[x] = \exp\left(-\frac{(x-\mu)^2}{2\sigma^2}\right), \tag{3}$$

where $\mu$ controls the center of the token budget in log-space (a larger $\mu$ prioritizes performance via more token requirement) and $\sigma$ regulates diversity in sampled configurations (a larger $\sigma$ explores extreme token budget). In practice, we generate $M$ configuration candidates by uniformly sweeping $\mu$ from 0 to $\log_2(b)$ with a fixed $\sigma$: $\mathcal{C} = \{\mathbf{p}^1, \mathbf{p}^2, \ldots, \mathbf{p}^M\}$ (see Appendix D.4 for a concrete example),

**Algorithm 1** Offline Pattern Determination

**Require:** Queries $\mathbf{Q} \in \mathbb{R}^{L \times d_h}$, keys $\mathbf{K} \in \mathbb{R}^{L \times d_h}$, configuration candidates $\mathcal{C} = \{\mathbf{p}^1, \ldots, \mathbf{p}^M\}$ threshold $\tau \in [0, 1]$.

1: Compute attention $\mathbf{A} = \text{softmax}\left(\mathbf{Q}\mathbf{K}^\top / \sqrt{d_h}\right)$
2: Initialize valid set $\mathcal{C}_{\text{valid}} \leftarrow \emptyset$
3: **for** each configuration $\mathbf{p}^i \in \mathcal{C}$ **do**
4:     Compute the selected token indexes $\mathcal{S}_i$ based on $\mathbf{p}^i$ as described in Section 4.2
5:     Compute aggregated score $\mathbf{a}_i$ via Eq. (4)
6:     **if** $\mathbf{a}_i \geq \tau$ **then** $\mathcal{C}_{\text{valid}} \leftarrow \mathcal{C}_{\text{valid}} \cup \{\mathbf{p}_i\}$
7: **end for**
8: $\mathbf{p}^* = \text{argmin}_{\mathbf{p}^i \in \mathcal{C}_{\text{valid}}} |\mathcal{S}_i|$

**Ensure:** Configuration $\mathbf{p}^*$

**Algorithm 2** Inference with Online Key Token Selection

**Require:** Input queries $\mathbf{Q} \in \mathbb{R}^{L \times d_h}$, keys $\mathbf{K} \in \mathbb{R}^{L \times d_h}$, values $\mathbf{V} \in \mathbb{R}^{L \times d_h}$, block size $b$, local window size $w$, configuration $\mathbf{p}^*$ generated by Alg. 1.

1: Compute token scores $\mathbf{s} = \text{softmax}(\mathbf{Q}_{L,:}\mathbf{K}^\top / \sqrt{d_h})$
2: Split $\mathbf{K}$ into $m = \lfloor L/b \rfloor$ blocks (same for $\mathbf{V}$)
3: Compute block sparsity $\mathbf{h} = [h_1, \ldots, h_m]$ via Eq. (6)
4: Sort blocks in descending order by the sparsity $\mathbf{h}$
5: Compute token requirement $\mathbf{t} = [t_1, t_2, \ldots, t_m]$ based on configuration $\mathbf{p}^*$ via Eq. (7)
6: Generated selected tokens index $\mathcal{S}$ via Eq. (8)
7: $\mathbf{Att} = \text{Softmax}\left(\mathbf{Q}[\mathbf{K}^G; \mathbf{K}^L]^\top / \sqrt{d_h}\right)[\mathbf{V}^G; \mathbf{V}^L]$, $\mathbf{K}^G = \mathbf{K}_\mathcal{S}, \mathbf{V}^G = \mathbf{V}_\mathcal{S}, \mathbf{K}^L = \mathbf{K}_{L-\mathbf{w}:,:}, \mathbf{V}^L = \mathbf{V}_{L-\mathbf{w}:,:}$

**Ensure:** Attention output $\mathbf{Att}$

where the $i$-th configuration $\mathbf{p}^i$ is computed with $\mu^i = \log_2\left(1 + \frac{(i-1) \cdot (b-1)}{M-1}\right)$. This ensures smooth interpolation between efficiency-oriented and accuracy-oriented sparsity patterns.

**Cumulative-Score Configuration Determination.** The goal of offline determination is to find the most efficient (sparsest) configuration from the candidate set $\mathcal{C}$ that maintains the head's performance. We measure performance via the aggregated attention score retained by the selected tokens. For a given candidate $\mathbf{p}^i$, we simulate its token selection on a calibration dataset (using our online mechanism from Sec. 4.2) to obtain the set of indices $\mathcal{S}_i$. We then compute the sum of the average attention scores for each selected token:

$$\mathbf{a}_i = \sum_{k \in \mathcal{S}_i} \frac{1}{\|\mathbf{A}_{:,k}\|_0} \sum_{j=1}^{\|\mathbf{A}_{:,k}\|_0} \mathbf{A}_{j,k}, \tag{4}$$

where $\mathbf{A} = \text{softmax}(\mathbf{Q}\mathbf{K}^\top / \sqrt{d_h}) \in \mathbb{R}^{L \times L}$ is computed on a calibration dataset, $\|\mathbf{A}_{:,k}\|_0$ denotes the number of non-zero elements in the $k$-th column of $\mathbf{A}$. We select the configuration $\mathbf{p}^* = \text{argmin}_{\mathbf{p}^i \in \mathcal{C}_{\text{valid}}} |\mathcal{S}_i|$, where $\mathcal{C}_{\text{valid}}$ denotes a configuration set with each $\mathbf{p}^i$ satisfying $\mathbf{a}_i \geq \tau$ and $|\mathcal{S}_i|$ denotes the number of preserved tokens. This selects the configuration that preserves the fewest tokens while retaining at least a threshold $\tau$ of the aggregated attention mass. This is a one-time, model-specific process, resulting in a task-agnostic configuration that generalizes across inputs.

## 4.2 ONLINE CORE CONTEXT SELECTION

During inference, the offline calibrated configuration $\mathbf{p}^*$ for each head dictates its overall token budget. The goal of the online stage is to distribute this budget adaptively across the input sequence based on the current context, thereby reducing the context-dependent redundancy. For an input sequence of length $L$, we first obtain a global, token-level importance score $\mathbf{s} \in \mathbb{R}^L$. Motivated by the observation that the final token has full visibility over all preceding tokens in the causal self-attention mechanism, we leverage this inherent property to identify tokens that are most relevant to the overall context. We compute the importance score by:

$$\mathbf{s} = \text{softmax}(\frac{\mathbf{Q}_{L,:}\mathbf{K}^\top}{\sqrt{d_h}}). \tag{5}$$

where $\mathbf{Q}_{L,:}$ is the query vector of the last token. Compared to methods like MInference and Flex-Prefill, which use the last $k$ tokens to score importance (with $\mathcal{O}(kL)$ complexity), This reduces the cost to $\mathcal{O}(L)$ while maintaining performance. This design is not only computationally efficient but also avoids introducing handcrafted heuristics or arbitrary hyperparameters (e.g., the choice of $k$). Then, similar to the offline phase, we partition the input sequence into non-overlapping blocks of size $b$. The tokens of the blocks that are not divisible are put into the local subset. For each block $\mathcal{B}_j$, we compute a redundancy score $\mathbf{h}_j$ that quantifies its information density:

$$\mathbf{h}_j = (1 - \alpha) \cdot \sum_{i \in \mathcal{B}_j} \mathbf{s}_i + \alpha \cdot \left(1 - \frac{\sum_{i \in \mathcal{B}_j} \mathbf{s}_i^2}{\left(\sum_{i \in \mathcal{B}_j} \mathbf{s}_i\right)^2}\right), \tag{6}$$

The first term (weighted by $1 - \alpha$) penalizes blocks with low total attention mass. The second term (weighted by $\alpha$) is a variant of the Herfindahl-Hirschman Index (Rhoades, 1993); it decreases as attention becomes more concentrated on a few tokens (indicating higher redundancy within the block). Thus, a lower $\mathbf{h}_j$ score indicates a more redundant block. The blocks are then sorted by their $\mathbf{h}_j$ scores, resulting in an ordered list of block indices $\mathbf{I} = \text{SortIndex}(\mathbf{h})$.

In online selection, we adaptively assign a token budget $\mathbf{t}_i$ to each block, dictated by its rank in $\mathbf{I}$ and the head's pre-defined configuration $\mathbf{p}^*$. Recall that $\mathbf{p}^* = [p_1, p_2, ..., p_b]$ defines the head's sparsity policy: it specifies that a proportion $p_k$ of blocks should be assigned a budget of $k$ tokens. We enforce this policy through a deterministic mapping:

$$\mathbf{t}_i = \Psi[\mathbf{I}_i], \text{ where } \Psi = \Big( \underbrace{1, \ldots, 1}_{\lfloor m \times p_1 \rfloor}, \underbrace{2, \ldots, 2}_{\lfloor m \times p_2 \rfloor}, \ldots, \underbrace{b \ldots, b}_{\lfloor m \times p_b \rfloor} \Big). \tag{7}$$

This mechanism ensures that the most information-dense blocks (highest rank in $\mathbf{I}$) receive the largest token budgets (later entries in $\Psi$), as dictated by the head's sparsity configuration $\mathbf{p}^*$. For each block $\mathcal{B}_i$, we select the top-$\mathbf{t}_i$ tokens with the highest importance scores $\mathbf{s}$:

$$\mathcal{S}_i = \text{Top}(\mathbf{t}_i; \mathcal{B}_i). \tag{8}$$

The union of all these subsets $\mathcal{S} = \bigcup_{i=1}^{m} \mathcal{S}_i$ forms the global subset. This set, concatenated with the local subset of the most recent $w$ tokens, is used to compute the final attention output via Eq. (2). As analyzed in Appendix E.9, the overhead of our lightweight redundancy metric is marginal compared to the significant computational savings achieved by sparsification.

**Online Token Selection in Decoding Stage**. Our method naturally extends to the decoding stage, where tokens are incrementally generated. Specifically, once the number of new tokens reaches the block size $b$, we evaluate the importance of its tokens using the same scoring function in Eq. (5) based on the last query. We then retain only the top-$t$ most informative tokens in the block based on their scores. The value $t = \lfloor \sum_{k \in \mathcal{K}} k \cdot p_k \rfloor$ corresponds to the average number of tokens preserved per block, derived from the determined configuration $\mathbf{p}^*$. This ensures that only the relevant context is kept, enabling efficient and effective long-sequence modeling in the decoding stage.

### 4.3 THEORETICAL GUARANTEES FOR TFCA-ATTENTION

To provide a theoretical foundation for our approach, we analyze the approximation error of TFCA-Attention relative to full attention. We show that the error is naturally bounded by the probability mass assigned to tokens not selected by our method, and is thus controllable via the threshold $\tau$.

**Theorem 1 (Error Bound for a Single Query).** *Let $\gamma_i = \sum_{j \notin \mathcal{S}_{total}} \frac{\exp(\mathbf{A}_{ij})}{Z_i}$ be the total probability mass assigned by the full softmax to tokens not selected by TFCA-Attention, where $Z_i$ is the normalization constant. Then the approximation error is bounded by*

$$|\mathbf{Att}_i - \widetilde{\mathbf{Att}}_i|_1 \leq 2\gamma_i \cdot |\mathbf{V}|_\infty, \tag{9}$$

*where $\mathbf{Att}_i$ and $\widetilde{\mathbf{Att}}_i$ denote the $i$-th rows of the full and approximate attention outputs, respectively, $|\mathbf{V}|_\infty$ is the maximum absolute value in the value matrix.*

Theorem 1 shows that the error scales linearly with $\gamma_i$, *i.e.*, the fraction of probability mass on unselected tokens. By configuring the token selection threshold $\tau$ to keep $\gamma_i$ small, TFCA-Attention explicitly controls the approximation error, ensuring faithfulness to the full attention mechanism.

## 5 EXPERIMENTS

We evaluate on two state-of-the-art LLMs: 1) LLaMA-3.1-8B-Instruct-128K (AI, 2024), 2) Qwen2.5-7B-Instruct-128K (Yang et al., 2024). For benchmarks, we consider long-context understanding tasks, *i.e.*, LongBench-E (Bai et al., 2023) and RULER (Hsieh et al., 2024), and short-context benchmarks. *i.e.*, MMLU (Hendrycks et al., 2021), GSM-8K (Cobbe et al., 2021a), and HumanEval (Chen et al., 2021). We set the threshold $\tau$, the block size $b$, the balancing parameter $\alpha$, and the window size $w$ to 0.9, 128, 0.5, and 4096, respectively. All the latency is measured on a single A800 GPU. All latency for full attention are based on the highly optimized FlashAttention-2 (Dao, 2024) kernel. See Appendix D for more details.

Table 2: Comparisons of different models on LongBench-E Bai et al. (2023). We report the attention computation latency in 64K context.

| Methods | S. QA | M. QA | Sum. | F. S. | Syn. | Code | Avg. | Latency (ms) |
|---|---|---|---|---|---|---|---|---|
| LLaMA3.1-8B-Instruct-128K | 51.63 | 53.23 | 30.78 | 68.67 | 54.29 | 60.52 | 53.19 | 316.14 |
| • MInference (Jiang et al., 2024) | 51.70 | 52.72 | 30.76 | 68.58 | 53.50 | 61.12 | 53.06 | 324.84 |
| • FlexPrefill (Lai et al., 2025) | 50.35 | 52.85 | 30.71 | 68.41 | 54.30 | 62.76 | 53.23 | 280.11 |
| • XAttention (Xu et al., 2025) | 49.96 | 51.98 | 31.22 | 68.07 | 48.50 | 55.75 | 50.91 | 133.85 |
| • TFCA-Attention (Ours) | 52.28 | 52.83 | 30.84 | 68.40 | 54.86 | 61.82 | **53.51** | 120.96 |
| Qwen2.5-7B-Instruct-128K | 48.75 | 52.24 | 27.81 | 65.00 | 52.00 | 61.14 | 51.16 | 268.55 |
| • MInference (Jiang et al., 2024) | 48.80 | 52.37 | 27.64 | 64.67 | 51.50 | 62.08 | 51.18 | 292.33 |
| • FlexPrefill (Lai et al., 2025) | 49.08 | 52.16 | 27.86 | 65.18 | 52.00 | 62.20 | 51.41 | 244.37 |
| • XAttention (Xu et al., 2025) | 48.50 | 50.08 | 27.48 | 66.40 | 50.50 | 60.98 | 50.66 | 119.87 |
| • TFCA-Attention (Ours) | 48.50 | 52.91 | 27.74 | 64.92 | 52.25 | 62.74 | **51.51** | 105.93 |

Table 3: Comparisons of different models on RULER. We report the latency in 128K context.

| Methods | 4K | 8K | 16K | 32K | 64K | 128K | Avg. | Latency (s) |
|---|---|---|---|---|---|---|---|---|
| LLaMA3.1-8B-Instruct-128K | 96.74 | 94.03 | 92.02 | 84.17 | 81.32 | 76.89 | 87.52 | 1.28 |
| • MInference (Jiang et al., 2024) | 96.54 | 94.06 | 91.37 | 85.79 | 83.03 | 54.13 | 84.15 | 0.84 |
| • FlexPrefill (Lai et al., 2025) | 95.99 | 93.67 | 92.73 | 88.14 | 81.14 | 74.67 | 87.72 | 1.02 |
| • XAttention (Xu et al., 2025) | 96.15 | 93.95 | 93.71 | 90.90 | 83.35 | 72.57 | 88.44 | 0.50 |
| • TFCA-Attention (Ours) | 96.31 | 95.38 | 93.92 | 86.38 | 82.89 | 77.46 | **88.72** | 0.45 |
| Qwen2.5-7B-Instruct-128K | 96.00 | 94.85 | 91.77 | 89.85 | 70.38 | 52.92 | 82.63 | 1.10 |
| • MInference (Jiang et al., 2024) | 96.08 | 94.92 | 91.69 | 89.92 | 70.46 | 51.62 | 82.45 | 0.77 |
| • FlexPrefill (Lai et al., 2025) | 95.62 | 94.31 | 92.00 | 88.23 | 70.23 | 52.15 | 82.09 | 0.89 |
| • XAttention (Xu et al., 2025) | 95.45 | 92.91 | 92.04 | 88.84 | 68.84 | 55.36 | 82.24 | 0.44 |
| • TFCA-Attention (Ours) | 96.00 | 94.77 | 91.92 | 90.08 | 69.85 | 52.31 | **82.49** | 0.40 |

## 5.1 COMPARISONS WITH STATE-OF-THE-ART METHODS

As a dynamic sparse attention method, TFCA-Attention is primarily compared with state-of-the-art sparse attention methods: MInference (Jiang et al., 2024), FlexPrefill (Lai et al., 2025), and XAttention (Xu et al., 2025). This constitutes a fair comparison within the same category of techniques. To address the broader impact on end-to-end inference, we further conduct experiments by combining the strongest prefilling-stage baselines with KV cache compression methods (see Table 5). We summarize differences from existing methods in Table 1.

Table 1: Characteristic comparisons between existing methods and our TFCA-Attention.

| Methods | Dynamic | Prefilling | Decoding |
|---|---|---|---|
| MInference | ✓ | ✓ | |
| FlexPrefill | ✓ | ✓ | |
| XAttention | ✓ | ✓ | |
| SnapKV | ✓ | | ✓ |
| CAKE | ✓ | | ✓ |
| **Ours** | ✓ | ✓ | ✓ |

**Comparisons on LongBench-E.** In Table 2, TFCA-Attention achieves the highest average score among sparse attention methods. For LLaMA3.1-8B, it attains a **2.6×** speedup (120.96ms vs. 316.14ms) at 64K context while outperforming XAttention by 2.6 average score. On Qwen2.5-7B, it delivers the best performance with the fastest latency (2.5× speedup). These results validate our method's effectiveness and architectural adaptability without compromising accuracy.

**Comparisons on RULER.** In Table 3, TFCA-Attention achieves superior efficiency and accuracy in long context understanding across different context lengths. On LLaMA3.1-8B-Instruct and Qwen2.5-7B-Instruct, TFCA-Attention attains the highest average score with up to **2.8×** speedup compared to vanilla self-attention (0.45s vs 1.28s), outperforming the strongest counterpart method XAttention despite faster computation. These results confirm that our dynamic sparsity mechanism effectively maintains performance while accelerating computation across varying sequence lengths.

**Comparisons on Short-context Tasks.** In Table 4, our TFCA-Attention achieves competitive results on LLaMA3.1-8B-Instruct-128K and outperforms all baselines on Qwen2.5-7B-Instruct-128K,

Table 4: Comparisons on short-context tasks, covering common sense, math, and code.

(a) Comparisons on LLaMA3.1-8B-Instruct.

| Methods | MMLU | GSM-8K | HumanEval |
|---|---|---|---|
| LLaMA3.1-8B-Instruct | **69.38** | 83.85 | **68.29** |
| • MInference | 69.14 | 84.08 | 67.30 |
| • FlexPrefill | 69.16 | 84.15 | 67.07 |
| • XAttention | 69.21 | 84.15 | 67.39 |
| • TFCA-Attention (Ours) | 69.21 | **84.23** | 67.46 |

(b) Comparisons on Qwen2.5-7B-Instruct.

| Methods | MMLU | GSM-8K | HumanEval |
|---|---|---|---|
| Qwen2.5-7B-Instruct | 74.22 | 79.68 | **81.71** |
| • MInference | 74.14 | 80.29 | 79.88 |
| • FlexPrefill | 74.23 | 80.36 | 81.10 |
| • XAttention | 74.20 | 79.30 | 80.49 |
| • TFCA-Attention (Ours) | **74.26** | 80.44 | 81.10 |

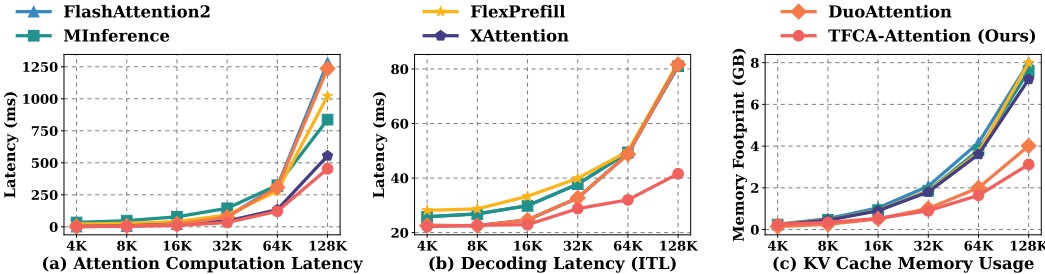

Figure 3: Comparisons in terms of computational and storage overhead on LLaMA3.1-8B-Instruct. Attention computation latency is the time to compute a single attention layer. "ITL" (inter token latency) is the time between generating consecutive tokens (except for the first token) in decoding.

Table 5: Performance comparisons across prefilling and decoding on LongBench-E.

| Methods | S. QA | M. QA | Sum. | F.S. | Syn. | Code | Avg. |
|---|---|---|---|---|---|---|---|
| MInference + SnapKV | 49.66 | 53.16 | 29.90 | 67.60 | 51.72 | 58.25 | 51.72 |
| FlexPrefill + CAKE | 50.27 | 52.87 | 30.77 | 66.96 | 52.25 | 60.24 | 52.23 |
| DuoAttention | 51.05 | 52.70 | 29.70 | 67.27 | 52.25 | 59.86 | 52.14 |
| TFCA-Attention (Ours) | 50.71 | 53.11 | 30.24 | 67.93 | 52.59 | 60.05 | **52.44** |

demonstrating robust cross-model generalization. Notably, it preserves the capability of the original model, validating that our method effectively retains critical information without fine-tuning.

**Comparisons on Computational Efficiency**. As shown in Figure 3, TFCA-Attention achieves a **2.8×** speedup in prefilling latency and a **2.1×** faster decoding speed over vanilla self-attention at 128K context length on LLaMA3.1-8B-Instruct. By dynamically selecting tokens, it also reduces KV cache memory by **61%** (3.12GB vs. 8.00GB). In contrast, methods like MInference only accelerate prefilling, leaving decoding latency and memory unchanged. Our analysis of the official DuoAttention implementation reveals that it does not accelerate the prefill stage, relying instead on a standard FlashAttention-2 kernel during this stage. Its optimization is primarily confined to KV cache compression and decoding. This demonstrates TFCA-Attention's unique advantage in enabling end-to-end efficiency.

**Performance Comparison across Prefilling and Decoding**. We further conduct experiments on LongBench-E to validate the effectiveness of our unified framework across *both prefilling and decoding stages*. We integrate the strongest methods from prefilling acceleration and KV cache compression to enable comprehensive benchmarking. The results in Table 5 demonstrate that our approach achieves superior performance across most tasks. This highlights that our method establishes a more efficient end-to-end framework for long-context processing.

## 5.2 Ablations

We perform ablation studies on Qwen2.5-7B-Instruct-128K and report the average score on LongBench-E and decoding latency in Figure 4 (see more results in Appendix E).

**Ablations on Block Size** $b$. As shown in Figure 4a, the model achieves the highest average score of 51.51 with a block size of 128. Smaller block sizes may fail to capture sufficient contextual redundancy, while larger sizes risk over-compressing critical information.

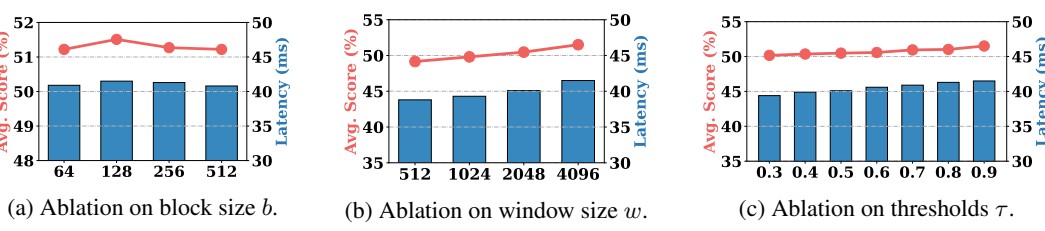

(a) Ablation on block size $b$.  (b) Ablation on window size $w$.  (c) Ablation on thresholds $\tau$.

Figure 4: Ablations on hyper-parameters. We report average score on LongBench-E and lantency.

**Ablations on Local Window Size** $w$. As shown in Figure 4b, performance peaks at $w = 4096$ (avg score 51.51), with latency increasing only marginally from 38.8ms to 41.5ms, indicating that preserving richer local context yields significant accuracy gains with minimal computational cost.

**Ablations on Threshold** $\tau$. In Figure 4c, increasing $\tau$ from 0.3 to 0.9 improves the average score, peaking at 51.51 for $\tau$=0.9, due to stricter retention of core context. Higher thresholds enhance context preservation by retaining more important tokens while incurring marginal latency increases.

## 6 CONCLUSION

In this work, we propose TFCA-Attention, a training-free dynamic attention mechanism that mitigates the critical bottleneck of quadratic complexity in long-context LLMs in a one-stone-three-birds manner: accelerating prefilling, reducing decoding latency, and compressing KV cache. By adapting to both head-specific and context-dependent redundancy, TFCA-Attention achieves efficient computation without sacrificing accuracy. Extensive experiments demonstrate that our approach provides significant efficiency gains (2.8× decoding speedup, 61% KV cache reduction) while maintaining full-attention accuracy. The plug-and-play nature of TFCA-Attention makes it a practical solution for deploying LLMs in long-context scenarios without retraining or architectural changes.

## REPRODUCIBILITY STATEMENT

To facilitate the reproducibility of our work, we have made the following efforts:

- **Datasets:** All datasets used in our experiments are publicly available benchmarks: LongBench-E (Bai et al., 2023), RULER (Hsieh et al., 2024), MMLU (Hendrycks et al., 2021), GSM-8K (Cobbe et al., 2021b), and HumanEval (Chen et al., 2021). We provide detailed descriptions of each dataset, including preprocessing steps and evaluation metrics, in Appendix D.2.
- **Implementation Details:** We provide a complete description of our method in Section 4, including pseudo-code in Algorithms 1 and 2. Additional implementation details, such as hyper-parameter settings, parallelization strategies, and Triton-based optimizations, are elaborated in Appendix D.3 and D.4. Our code will be released upon acceptance.
- **Computational Resources:** All experiments were conducted on a single node with 8 NVIDIA A800 GPUs (80GB memory each). We report latency and memory usage for both prefill and decoding stages in Section 5, and provide a detailed latency breakdown in Appendix E.9.
- **Theoretical Claims:** We include a full theoretical analysis of the approximation error bound of TFCA-Attention in Appendix A, with proofs clearly stated.

We believe these efforts will enable researchers to reproduce our results and build upon our work.

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

# APPENDIX

## CONTENTS

# A  THEORETICAL ANALYSIS OF APPROXIMATION ERROR

A central question for any sparse attention mechanism is its capacity to accurately approximate the full attention output. In this section, we present a theoretical analysis that bounds the approximation error of TFCA-Attention. We show that the error is bounded and can be explicitly controlled via the method's hyperparameters, particularly the threshold $\tau$ used in the offline configuration process.

For a single attention head, let the full attention output be $\mathbf{Att} = \text{softmax}(\mathbf{A})\mathbf{V}$, where $\mathbf{A} = \frac{\mathbf{QK}^\top}{\sqrt{d_h}} \in \mathbb{R}^{L \times L}$ is the pre-softmax score matrix. TFCA-Attention computes an approximation $\widetilde{\mathbf{Att}} = \text{softmax}(\widetilde{\mathbf{A}})\widetilde{\mathbf{V}}$, where $\widetilde{\mathbf{A}} = \frac{\mathbf{Q}\widetilde{\mathbf{K}}^\top}{\sqrt{d_h}}$, and $\widetilde{\mathbf{K}}, \widetilde{\mathbf{V}} \in \mathbb{R}^{L \times d_h}$ contain only the selected global and local tokens (with other rows zeroed). The token selection is determined by the index set $\mathcal{S}_{\text{total}} = \mathcal{S} \cup \mathcal{S}_{\text{local}}$. We measure the error for a single query vector $\mathbf{q}_i$ (the $i$-th row of $\mathbf{Q}$) in terms of the $\ell_1$ norm between the full attention output vector and our approximation.

**Theorem 1 (Error Bound for a Single Query)** *Let $\gamma_i = \sum_{j \notin \mathcal{S}_{total}} \frac{\exp(\mathbf{A}_{ij})}{Z_i}$ be the total probability mass assigned by the full softmax to tokens not selected by TFCA-Attention, where $Z_i$ is the normalization constant. Then the approximation error is bounded by*

$$|\mathbf{Att}_i - \widetilde{\mathbf{Att}}_i|_1 \leq 2\gamma_i \cdot |\mathbf{V}|_\infty, \tag{10}$$

*where $\mathbf{Att}_i$ and $\widetilde{\mathbf{Att}}_i$ denote the $i$-th rows of the full and approximate attention outputs, respectively, $|\mathbf{V}|_\infty$ is the maximum absolute value in the value matrix.*

*Proof.* Let $\mathbf{s}_i = \text{softmax}(\mathbf{A}_i) = \frac{\exp(\mathbf{A}_i)}{Z_i}$ be the full attention probability distribution for query $i$. Let $\widetilde{\mathbf{s}}_i = \text{softmax}(\widetilde{\mathbf{A}}_i)$ be the approximate distribution. Note that $\widetilde{\mathbf{s}}_i$ is defined only over the selected tokens; for mathematical convenience, we consider it a distribution over all $L$ tokens by setting $(\widetilde{\mathbf{s}}_i)_j = 0$ for $j \notin \mathcal{S}_{\text{total}}$. Then, the attention outputs are:

$$\mathbf{Att}_i = \mathbf{s}_i \mathbf{V}, \quad \widetilde{\mathbf{Att}}_i = \widetilde{\mathbf{s}}_i \widetilde{\mathbf{V}}$$

Note that $\widetilde{\mathbf{V}}$ is $\mathbf{V}$ with rows zeroed out for unselected tokens, so $\widetilde{\mathbf{V}} = \mathbf{M} \odot \mathbf{V}$, where $\mathbf{M}$ is a masking matrix. Thus, the error can be split using the triangle inequality:

$$E_i = |\mathbf{Att}_i - \widetilde{\mathbf{Att}}_i|_1 = \|\mathbf{s}_i \mathbf{V} - \widetilde{\mathbf{s}}_i \widetilde{\mathbf{V}}\|_1 \leq \underbrace{\|(\mathbf{s}_i - \widetilde{\mathbf{s}}_i)\mathbf{V}\|_1}_{\text{Term I: Softmax Error}} + \underbrace{\|\widetilde{\mathbf{s}}_i(\mathbf{V} - \widetilde{\mathbf{V}})\|_1}_{\text{Term II: Value Error}}. \tag{11}$$

**1) Bounding Term II (Value Error)**: Since $\widetilde{\mathbf{V}}$ and $\mathbf{V}$ are identical for all selected tokens $j \in \mathcal{S}_{\text{total}}$, and $\widetilde{\mathbf{s}}_i$ is zero for unselected tokens, this term is zero.

$$\text{Term II} = \|\widetilde{\mathbf{s}}_i(\mathbf{V} - \widetilde{\mathbf{V}})\|_1 = 0. \tag{12}$$

**2) Bounding Term I (Softmax Error)**: We now bound the difference between the two distributions $\mathbf{s}_i$ and $\widetilde{\mathbf{s}}_i$.

$$\text{Term I} = \|(\mathbf{s}_i - \widetilde{\mathbf{s}}_i)\mathbf{V}\|_1 \leq \|\mathbf{s}_i - \widetilde{\mathbf{s}}_i\|_1 \cdot \|\mathbf{V}\|_\infty. \tag{13}$$

We directly analyze the total variation. Let $Z_i = \sum_j \exp(A_{ij})$ and $\widetilde{Z}_i = \sum_{j \in \mathcal{S}_{\text{total}}} \exp(A_{ij})$. The difference is:

$$\|\mathbf{s}_i - \widetilde{\mathbf{s}}_i\|_1 = \sum_{j \in \mathcal{S}_{\text{total}}} \left| \frac{\exp(A_{ij})}{Z_i} - \frac{\exp(A_{ij})}{\widetilde{Z}_i} \right| + \sum_{j \notin \mathcal{S}_{\text{total}}} \frac{\exp(A_{ij})}{Z_i}$$

The second term is precisely $\gamma_i$ by definition. The first term can be simplified:

$$\sum_{j \in \mathcal{S}_{\text{total}}} \exp(A_{ij}) \left| \frac{1}{Z_i} - \frac{1}{\widetilde{Z}_i} \right| = \widetilde{Z}_i \cdot \frac{|Z_i - \widetilde{Z}_i|}{Z_i \widetilde{Z}_i} = \frac{|Z_i - \widetilde{Z}_i|}{Z_i}$$

Recall that $\gamma_i = \sum_{j \notin \mathcal{S}_{\text{total}}} \frac{\exp(A_{ij})}{Z_i} = \frac{Z_i - \widetilde{Z}_i}{Z_i}$. Thus, we have

$$\|\mathbf{s}_i - \widetilde{\mathbf{s}}_i\|_1 = \sum_{j \in \mathcal{S}_{\text{total}}} \left| \frac{\exp(A_{ij})}{Z_i} - \frac{\exp(A_{ij})}{\widetilde{Z}_i} \right| + \gamma_i$$

$$= \gamma_i \left( \frac{\widetilde{Z}_i}{Z_i} + 1 \right) = \gamma_i \left( (1 - \gamma_i) + 1 \right) = \gamma_i (2 - \gamma_i) \leq 2\gamma_i.$$

Since $\gamma_i \in [0, 1]$, the term $\gamma_i(2 - \gamma_i)$ is always $\leq 2\gamma_i$. This is a tighter bound. Therefore, we get

$$\text{Term I} \leq 2\gamma_i \cdot \|\mathbf{V}\|_\infty. \tag{14}$$

According to the bounds for Term I in Eqn. (14) and Term II in Eqn. (12), we obtain the result:

$$E_i \leq 2\gamma_i \cdot \|\mathbf{V}\|_\infty. \tag{15}$$

$\square$

**Interpretation and Discussion.** Theorem 1 provides a rigorous bound on the approximation error for each query position. The error scales linearly with $\gamma_i$, the total probability mass of the full attention is assigned to the tokens discarded by TFCA-Attention.

- **Controllability:** Our offline calibration phase (Section 4.1) directly controls this error. By setting a threshold $\tau$ (e.g., 0.9) and ensuring that the aggregated attention mass of the selected tokens $\mathbf{a}_i \geq \tau$, we are effectively enforcing that $\gamma_i \leq 1 - \tau$ for the calibration data. This guarantees that the error bound is approximately limited to $(1-\tau) \cdot \|\mathbf{V}\|_\infty$ on the calibration distribution. By using a small calibration set, we ensure this bound generalizes to test inputs, which is demonstrated in Appendix E.6.

- **Role of Local Context:** The local window $\mathcal{S}_{\text{local}}$ ensures that the most recent tokens, which often have high attention scores for generative tasks, are always included. This prevents $\gamma_i$ from becoming large for these critical query positions, thus proactively minimizing the error bound.

- **Head-Specificity:** The error bound is per-head and per-query. This aligns with our design: heads with higher inherent redundancy (lower $\gamma_i$ for the same number of discarded tokens) can tolerate a sparser configuration ($\mathbf{p}^*$ with a lower budget) without violating the error constraint.

In conclusion, TFCA-Attention is a principled approximation algorithm whose output deviation from full attention is theoretically bounded and explicitly managed by its configuration process. The empirical results showing maintained performance across diverse benchmarks (Section 5) provide strong evidence that the values of $\gamma_i$ encountered in practice are indeed small, validating our theoretical analysis.

# B MORE VISUALIZATIONS OF ATTENTION SCORE DISTRIBUTIONS

In this section, we provide more visualizations of attention score distributions for both LLaMA-3.1-8B-Instruct (Figure 5) and Qwen2.5-7B-Instruct (Figure 6). Our analysis reveals consistent patterns across models: 1) significant variation in attention distributions across different layers and heads; 2) position-dependent scoring patterns throughout the sequence length; 3) the emergence of all four distribution types identified in Section 3.2 (uniform, bipolar, terminal bias, attention sink, and sparse activation). These observations hold regardless of model architecture, strongly motivating our design of a unified efficient attention mechanism that can automatically adapt to these heterogeneous patterns while maintaining computational efficiency. The cross-model consistency of these findings suggests that our approach may generalize well to other transformer-based LLMs.

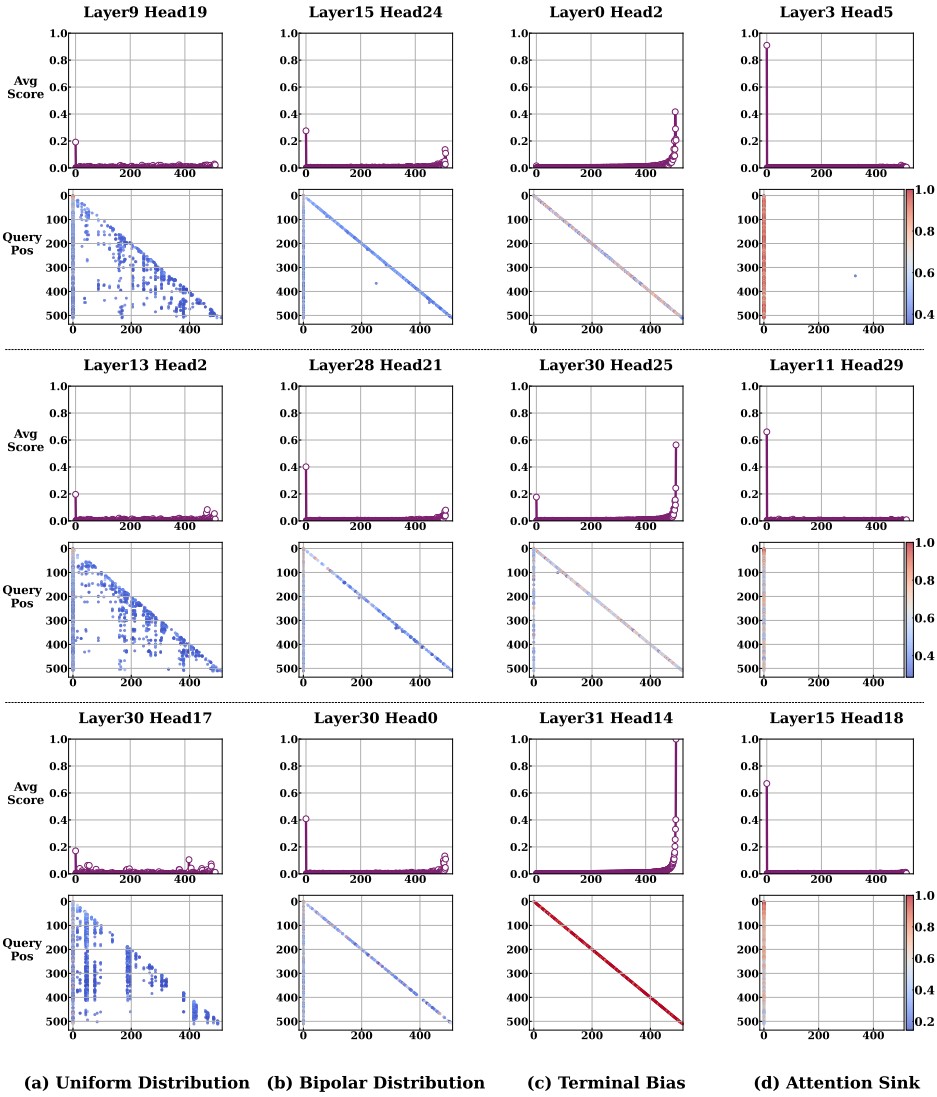

Figure 5: More visualization of attention score distributions across different layers and heads in LLaMA-3.1-8B-Instruct.

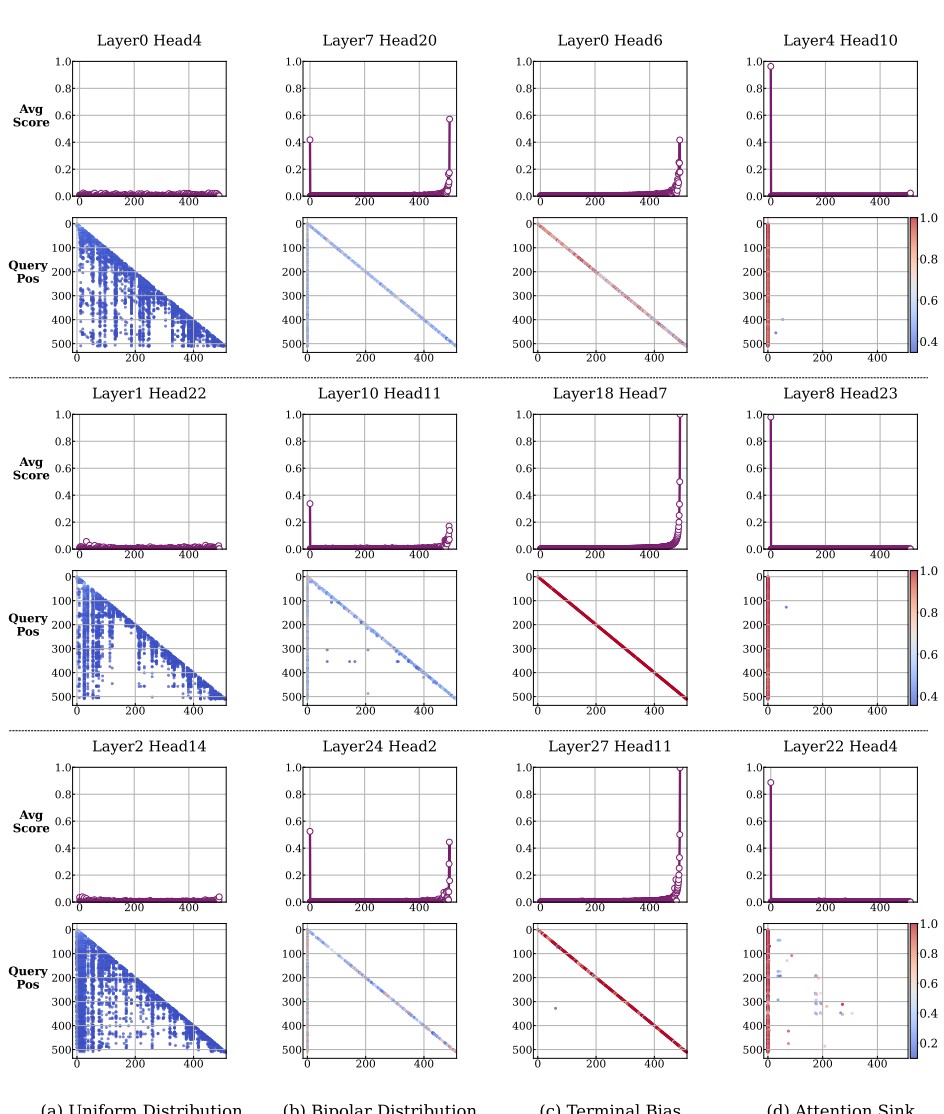

Figure 6: More visualization of attention score distributions across different layers and heads in Qwen2.5-7B-Instruct. (continued from previous page)

# C    MORE DISCUSSIONS

## C.1    MORE DISCUSSIONS ON DIFFERENCES FROM EXISTING METHODS

Self-attention mechanisms (Vaswani et al., 2017) in LLMs face significant computational redundancy when processing ultra-long contexts (e.g., 128K tokens). As identified in our Section 1, the redundancy in attention distributions exhibits two critical properties: (1) head-specific redundancy, where different attention heads exhibit diverse sparsity patterns across layers, and (2) context-dependent redundancy, where token importance varies dynamically with input content. While these properties have been partially observed in prior work, no existing method adequately addresses both in a unified manner across prefilling and decoding stages. Based on these observations, we propose TFCA-Attention, which simultaneously adapts to both types of redundancy while accelerating both prefilling and decoding stages. This section further clarifies our methodological distinctions from recent dynamic sparse attention methods and KV cache compression methods:

**Unified Acceleration of Prefilling and Decoding.** Existing approaches are typically limited to optimizing either prefilling or decoding due to fundamental architectural constraints. Sparse attention methods like Minference (Jiang et al., 2024) and FlexPrefill (Lai et al., 2025) accelerate prefilling by computing subsets of attention scores through predefined patterns. However, these methods rely on predefined sparse patterns and 2D attention block scoring. These techniques require computing a 2D attention map, which is infeasible under sequential decoding constraints. KV cache compression techniques such as SnapKV (Li et al., 2024) reduce memory during decoding by evicting or merging tokens, but operate after full attention computation and thus cannot accelerate the prefilling stage.

TFCA-Attention achieves unified acceleration through a token selection mechanism designed for both computational stages. Global redundancy is removed before prefilling attention computaion, while local context is preserved—accelerating computation without sacrificing quality. In decoding stage, Block-wise token selection is position-aware, metric-driven, and unbound by 2D dependencies, making it suitable for sequential generation. This explains our superior latency in Figure 3.

**Adaptation to Head-Specific and Context-Dependent Redundancy.** Most existing methods fail to adequately address both of head-specific and context-dependent redundancy due to their design paradigms. Fixed-pattern sparse attention methods (Zaheer et al., 2020; Beltagy et al., 2020; Xiao et al., 2024) apply uniform patterns across all heads, ignoring variations in redundancy distributions. While dynamic variants (Jiang et al., 2024; Xu et al., 2025; Lai et al., 2025) adapt per head, they select from a limited set of hand-designed patterns that cannot capture intricate redundancy variation. KV cache compression methods Hao et al. (2025); Wan et al. (2025); Qin et al. (2025); Li et al. (2024) typically apply uniform policies across all heads within a layer, neglecting head-specific characteristics.

TFCA-Attention addresses both redundancies through a decoupled two-phase approach. The offline configuration phase estimates each head's intrinsic redundancy level to determine head-specific token budgets. The online selection phase then identifies core tokens based on local contextual importance using efficiently computable metrics. This design enables adaptation to both head-specific sparsity characteristics and context-dependent token importance.

In summary, TFCA-Attention differs from prior work by simultaneously addressing both identified forms of redundancy through a unified method that accelerates both inference stages while maintaining comparable performance with vanilla self-attention.

## C.2    MORE DISCUSSIONS ON SCALABILITY

**Context length**: As shown in Figure 3, TFCA-Attention's efficiency gains increase with longer contexts. This is because longer sequences naturally contain more redundancy, making our dynamic token selection increasingly beneficial. The speedup ratio grows from $2.6\times$ at 64K to $2.8\times$ at 128K context length.

**Number of heads**: Our method demonstrates consistent effectiveness across models with different head counts. We validated TFCA-Attention on both LLaMA3.1-8B (32 heads) and Qwen2.5-7B (28 heads), showing that TFCA-Attention works regardless of the specific head count.

**Batch size**: We follow the standard practice in long-context acceleration research (MInference (Jiang et al., 2024), FlexPrefill (Lai et al., 2025), XAttention (Xu et al., 2025)) by evaluating latency at batch size=1, where sequence length (L), not batch size, dominates computational bottlenecks. While our current implementation targets the predominant long-context scenario, the theoretical overhead would scale sublinearly due to shared redundancy pattern determination across batches. We left this in the future.

# D  MORE IMPLEMENTATION DETAILS

## D.1  MORE DETAILS ON MODELS

We employ two state-of-the-art LLMs renowned for their exceptional performance in long-context tasks: 1) LLaMA-3.1-8B-Instruct-128K (AI, 2024), 2) Qwen2.5-7B-Instruct-128K (Yang et al., 2024). All selected models are instruction-tuned for chat-based interactions. We use the default chat templates provided with each model in the experiments.

## D.2  MORE DETAILS ON DATASET AND EVALUATION METRICS

**LongBench (Bai et al., 2023)** represents a cutting-edge benchmark architecture engineered for the systematic assessment of LLMs across three critical dimensions: bilingual (Chinese-English) competence, multitask generalization, and long-context semantic processing. Its cross-linguistic design enables rigorous comparative analysis of multilingual contextual encoding abilities in scenarios demanding comprehension of extended textual sequences exceeding standard input boundaries. Structured into six overarching task categories—encompassing single-document question answering, multi-document reasoning, summarization, few-shot learning paradigms, synthetic linguistic tasks, and code completion workflows—the benchmark instantiates 21 meticulously designed subtasks that span the core application domains of long-text processing. Specifically, the corpus includes 14 English-language tasks, 5 Chinese-language tasks, and 2 code-oriented evaluation modules, with median sequence lengths ranging from 5K to 15K tokens and a total of 4,750 instances. Additionally, LongBench-E is an enhanced variant of the benchmark, specifically crafted to assess model performance across input lengths on English tasks.

**RULER (Hsieh et al., 2024)** is a next-generation synthetic benchmark based on the NIAH paradigm, designed to evaluate the long-context capabilities of language model through configurable task complexity and sequence length. It extends traditional "needle" concepts into a taxonomy of semantic entities, relational patterns, and structural anomalies, enabling adjustable needle density to assess hierarchical information processing. The framework includes 13 subtasks across four categories: 8 retrieval tasks testing exact/semantic retrieval under noise, 3 multi-hop tracing tasks assessing sequential reasoning, 1 aggregation task evaluating contextual integration, and 1 complex QA task simulating multi-step inference. Its synthetic data pipeline ensures precise control over context length and needle correlations, supporting rigorous ablation studies and performance analysis of sequential memory, structural understanding, and multi-step reasoning.

**Massive Multitask Language Understanding (MMLU) (Hendrycks et al., 2021)** dataset is a comprehensive, multi-domain benchmark designed to rigorously assess the intellectual breadth and reasoning capabilities of language models across 57 diverse subject areas. These subjects span a wide spectrum of disciplines, including STEM, humanities, social sciences, and professional fields such as law and medicine, challenging models to demonstrate both factual knowledge and higher-order cognitive skills like analysis, inference, and application. Each question is crafted as a multiple-choice problem that requires nuanced understanding and logical reasoning, often surpassing surface-level pattern matching to evaluate a model's ability to generalize knowledge in zero-shot and few-shot settings. By incorporating content ranging from elementary concepts to advanced expertise, MMLU simulates real-world scenarios where models must rely on their pretraining knowledge without task-specific fine-tuning. This interdisciplinary design makes MMLU a cornerstone for evaluating the adaptability, knowledge retention, and cross-domain generalization of modern language models, offering critical insights into their performance in realistic, knowledge-intensive environments.

**GSM8K (Cobbe et al., 2021a)** is a benchmark designed to evaluate language models' numerical reasoning and problem-solving abilities through grade-school math problems, consisting of 7,473

training and 1,319 test samples that focus on multi-step reasoning tasks involving 2 to 8 sequential steps, requiring models to break down complex problems into logical sub-tasks while demonstrating both conceptual understanding of mathematical principles like arithmetic, proportions, and algebra and precise computation skills. The dataset's handcrafted problems exhibit rich linguistic variability with diverse narrative structures and real-world contextualizations, ensuring that models interpret nuanced language alongside solving equations, and it emphasizes chain-of-thought reasoning by encouraging models to articulate intermediate steps rather than provide direct answers, offering deeper insights into their logical processes and error patterns.

**HumanEval** (Chen et al., 2021) is a carefully curated code generation benchmark that consists of 164 hand-crafted Python programming problems, each designed to rigorously assess the coding capabilities of language models. Each problem in the dataset includes detailed specifications such as function signatures, docstrings, and a set of test cases, providing a comprehensive framework to evaluate both the syntactic correctness and functional accuracy of generated code. By covering a wide range of programming tasks—from basic algorithmic challenges to more complex logic-based problems—HumanEval ensures a thorough assessment of a model's ability to produce high-quality, executable code. The inclusion of unit tests further allows for precise validation of functional correctness, simulating real-world software development scenarios. As a result, HumanEval has become a cornerstone for evaluating the coding proficiency, logical reasoning, and problem-solving abilities of large language models in practical programming contexts.

Table 6: The candidate configurations used in offline sparsity determination.

| $\mu$ | 1 | 2 | 4 | 8 | 16 | 32 | 64 | 128 |
|---|---|---|---|---|---|---|---|---|
| 0.00 | 33.26% | 29.36% | 20.18% | 10.80% | 4.50% | 1.46% | 0.37% | 0.07% |
| 0.58 | 26.99% | 27.57% | 21.94% | 13.59% | 6.56% | 2.46% | 0.72% | 0.17% |
| 1.00 | 22.71% | 25.73% | 22.71% | 15.61% | 8.35% | 3.48% | 1.13% | 0.28% |
| 1.58 | 17.09% | 22.42% | 22.90% | 18.22% | 11.29% | 5.45% | 2.05% | 0.58% |
| 2.00 | 13.53% | 19.69% | 22.31% | 19.69% | 13.53% | 7.24% | 3.02% | 0.99% |
| 2.58 | 9.26% | 15.60% | 20.46% | 20.90% | 16.63% | 10.30% | 4.97% | 1.88% |
| 3.00 | 6.82% | 12.74% | 18.53% | 21.00% | 18.53% | 12.74% | 6.82% | 2.82% |
| 3.58 | 4.18% | 9.05% | 15.23% | 19.98% | 20.41% | 16.24% | 10.06% | 4.85% |
| 4.00 | 2.84% | 6.82% | 12.74% | 18.53% | 21.00% | 18.53% | 12.74% | 6.80% |
| 4.58 | 1.56% | 4.33% | 9.36% | 15.76% | 20.67% | 21.12% | 16.80% | 10.40% |
| 5.00 | 0.98% | 3.02% | 7.24% | 13.53% | 19.69% | 22.31% | 19.69% | 13.54% |
| 5.58 | 0.49% | 1.73% | 4.81% | 10.40% | 17.51% | 22.96% | 23.45% | 18.65% |
| 6.00 | 0.29% | 1.13% | 3.48% | 8.35% | 15.61% | 22.71% | 25.73% | 22.70% |
| 6.58 | 0.13% | 0.60% | 2.13% | 5.90% | 12.76% | 21.49% | 28.19% | 28.80% |

D.3 MORE DETAILS ON PARALLEL IMPLEMENTATION

To address the challenges of efficient parallelization in TFCA-Attention, we have developed specialized implementation strategies that overcome two critical barriers: non-contiguous memory access patterns and variable-length attention computation across attention heads.

**Memory Access Optimization**. The top-k token selection would lead to non-contiguous memory access, significantly degrading GPU performance. To mitigate this, we employ a parallel tokens gathering strategy that groups blocks where blocks within each group share the same top-k value. For each group, we perform the top-k selection in parallel across its blocks. Crucially, token selection is implemented by loading contiguous chunks of tokens within each block (since tokens are stored contiguously per block) and then gathering only the selected tokens. The selected tokens from all blocks are concatenated to form the global subsets $\mathbf{K}^G, \mathbf{V}^G$. This grouping strategy reduces memory fragmentation by minimizing random access scope, achieving near-optimal memory bandwidth utilization.

**Multi-head Parallelization**. Since each attention head requires a different number of tokens, this presents a significant challenge for parallelization. To address this, we flatten all head tokens into a single contiguous buffer and precompute offsets that map to each head's token region. During Triton kernel execution, each parallel thread uses these offsets to directly access the corresponding token region for its assigned head. This approach enables efficient parallel processing of variable-

length multi-head attention computation without synchronization overhead, making full use of GPU parallelism while maintaining the head-aware design principle.

**Future Optimization Directions**. We plan to further enhance performance through latency-hiding scheduling techniques. Specifically, while computing attention between queries and the local subset $\mathbf{K}^L, \mathbf{V}^L$, we can concurrently perform global token selection in the background. Once local attention computation completes, the global subsets $\mathbf{K}^G, \mathbf{V}^G$ would be ready, allowing seamless transition to global attention without additional delay. While this approach shows promise for further reducing end-to-end latency, it requires sophisticated kernel-level scheduling and is left as future work.

### D.4 MORE EXPERIMENTAL PROTOCOLS

**TFCA-Attention (Ours)**. We integrate the proposed TFCA-Attention with existing LLMs through a **plug-and-play** replacement of full self-attention, requiring **no architectural modifications** or **parameter updates**. Our experiments are conducted on a computational node equipped with 8 NVIDIA A800 GPUs, each with 80GB of memory. The experiments pipeline is implemented using PyTorch. Following FlashAttention Dao (2024), we further optimize the computation process of TFCA-Attention through Triton Tillet et al. (2019). The implementation follows GPU-aligned memory access patterns and cache-friendly computation schemes to maximize hardware utilization, while the dynamic token selection inherently eliminates redundant computations.

The threshold $\tau$ in offline determination is set to 0.9. The block size $b$ and window size $w$ are set to 128 and 4096 for all models, respectively. In all experiments, we adopt the 14 candidate configurations shown in Table 6 for all LLMs, where the appropriate configuration for each head is determined through Offline Sparsity Pattern Determination in Section 4.1. These candidate configurations are generated using Eq. 3. $\sigma$ is set to 2 and $\mu$ is sampled uniformly from $\log_2(1)$ to $\log_2(b)$ with 14 discrete steps. We find that setting $\sigma$ worked well in our experiments, so we did not further tune this hyperparameter. To ensure fair comparison with existing methods, we follow the standard practice in long-context acceleration research (Jiang et al., 2024; Lai et al., 2025; Xu et al., 2025) by evaluating latency at batch size=1. All experiments employed a greedy decoding strategy to ensure reproducibility and eliminate sampling variance.

**Compared Methods**. We compare our approach against three state-of-the-art baseline methods: **1) Minference** Jiang et al. (2024): This method employs offline determination to select optimal sparse attention patterns per attention head, combined with online dynamic adjustment of computation regions for each pattern. We use the officially released implementation in our all experiments. The pattern determined for all attention heads in the evaluated LLMs are vertical-slash patterns. Specifically, these patterns predominantly consisted of 1,000 vertical lines and 6,096 slash lines across different heads and layers. **2) FlexPrefill** Lai et al. (2025): This approach dynamically selects between Query-Aware and Vertical-Slash attention patterns per head, while adaptively determining the required Key-Value indices for computation. In our experiments, we use the official implementation and follow the original paper's settings: $\gamma$=0.9, $\tau$=0.1, with a minimum retained budget of 1,024 tokens and a block size of 128 across all evaluated models. **2) XAttention** Xu et al. (2025): This method employs an antidiagonal scoring pattern to select sparse attention blocks, reducing computation by processing only the selected regions. In our experiments, we follow the original paper's optimal setting with stride $S = 8$. For LLaMA3.1-8B, we adopt the officially released set of minimum thresholds to determine block selection. For Qwen2.5-7B, we set the threshold to 0.9 – higher than the paper's average recommendation of 0.8 – to preserve more contextual information.

### E MORE EXPERIMENTAL RESULTS

### E.1 COMPARISONS ON REASONING BENCHMARK

To further validate TFCA-Attention's effectiveness on highly challenging reasoning tasks, we evaluate our method using Qwen2.5-7B-Instruct on OlympiadBench, a challenging Olympiad-level benchmark covering math and physics reasoning. As shown in Table 7, TFCA-Attention demonstrates exceptional performance while significantly accelerating inference. Notably, TFCA-Attention matches the baseline model's mathematics performance while actually improving physics accuracy (19.95% vs. 19.73%), resulting in a slightly higher overall score (31.27% vs. 31.20%).

This is particularly impressive given that TFCA-Attention achieves these results with $2.8\times$ faster inference compared to the vanilla self-attention. The results confirm that TFCA-Attention effectively preserves critical reasoning paths even for highly complex tasks that require sophisticated multi-step reasoning, validating the ability to maintain performance while substantially improving efficiency for the most demanding cognitive tasks.

Table 7: Results on OlympiadBench with Qwen2.5-7B-Instruct and attention computation latency at 128K context.

| Model | Math | Physics | Avg. | Latency (s)↓ |
|---|---|---|---|---|
| Qwen2.5-7B-Instruct | **38.85** | 19.73 | 31.20 | 1.10 |
| ● FlexPrefill | 38.73 | 19.06 | 30.86 | 0.44 |
| ● TFCA-Attention (Ours) | 38.82 | **19.95** | **31.27** | **0.40** |

## E.2 COMPARISONS ON MULTI-TURN CONVERSATION BENCHMARK

Based on the comprehensive evaluation on the multi-turn conversation benchmark MT-Bench-101, our proposed TFCA-Attention demonstrates superior performance in conversational settings. As shown in Table 8, TFCA-Attention achieves an average score of 8.97, outperforming both the full-attention baseline (8.90) and the strongest competitor FlexPrefill (8.90). This performance advantage is consistent across multiple dialogue dimensions including Generation (GR), Reasoning (CR), and Safety (SA), where our method shows particularly strong results. These findings conclusively demonstrate that our token selection strategy, while computationally efficient, effectively preserves critical conversational context and reasoning paths across multiple dialogue turns. The results validate TFCA-Attention's robustness in real-world conversational applications while maintaining its training-free, plug-and-play advantage over methods that require architectural modifications or parameter updates.

Table 8: Results on MT-Bench-101 with Qwen2.5-7B-Instruct.

| Methods | GR | IC | AR | FR | MR | CC | TS | CR | SA | SI | CM | PI | SC | average |
|---|---|---|---|---|---|---|---|---|---|---|---|---|---|---|
| Qwen2.5-7B-Instruct | 8.20 | **7.71** | 9.49 | 9.56 | **7.57** | 9.89 | 8.90 | 9.50 | **9.29** | 8.26 | 9.19 | **8.64** | 9.42 | 8.90 |
| ● FlexPrefill | 7.81 | 7.67 | 9.55 | 9.57 | 7.23 | 9.91 | **9.40** | **9.57** | 9.22 | 8.33 | **9.27** | 8.63 | **9.53** | 8.90 |
| ● TFCA-Attention (Ours) | **8.50** | **7.71** | **9.56** | **9.74** | 7.48 | **9.91** | 9.39 | 9.49 | 9.26 | **8.38** | 9.20 | 8.62 | 9.43 | **8.97** |

## E.3 COMPARISONS WITH TRAINING-BASED METHODS

We compare our TFCA-Attention with training-based methods on LongBench-E, including CCA-Attention (Chen et al., 2025). In Table 9, TFCA-Attention achieves performance on par with the original model (22.28 vs. 22.42 average score), without any architectural modifications or training overhead. In contrast, CCA-Attention involves parameter updates, incurring high computational costs. In comparison, TFCA-Attention enables *plug-and-play deployment*.

Table 9: Comparisons with training-based methods on LongBench-E Bai et al. (2023).

| Methods | Training-free | S. QA | M. QA | Sum. | F.S. | Syn. | Code | Avg. |
|---|---|---|---|---|---|---|---|---|
| LLaMA2-7B-Instruct-80K | – | 3.22 | 2.71 | 3.90 | 64.98 | 0.56 | 59.16 | 22.42 |
| ● CCA-LLM (Chen et al., 2025) | ✗ | 5.62 | 4.34 | 8.99 | 59.60 | 0.48 | 54.40 | 22.24 |
| ● TFCA-Attention (Ours) | ✓ | 3.38 | 2.72 | 4.01 | 63.39 | 0.64 | 59.52 | 22.28 |

## E.4 EFFECTIVENESS OF LOCAL-CONTEXT REDUNDANCY METRIC

To validate the efficacy of our proposed *Local-context Redundancy Metric* $\mathbf{h}$ for redundancy estimation, we conduct an ablation study comparing our Online Core Context Selection strategy with a variant that randomly sorts blocks on LongBench-E with Qwen2.5-7B-Instruct. As shown in Table 10, our metric-driven approach achieves **1.59%** average score improvement over the random baseline. This significant performance gap demonstrates that the proposed metric accurately quantifies block-wise redundancy levels and enables adaptive token retention requirements.

Table 10: Comparisons of different models on LongBench-E Bai et al. (2023).

| Methods | S. QA | M. QA | Sum. | F.S. | Syn. | Code | Avg. |
|---------|-------|-------|------|------|------|------|------|
| Qwen2.5-7B-Instruct-128K | 48.75 | 52.24 | 27.81 | 65.00 | 52.00 | 61.14 | 51.16 |
| ● Random Sort | 47.19 | 50.78 | 26.63 | 63.35 | 51.50 | 60.07 | 49.92 |
| ● Sort with Metric $\mathbf{h}$ | 48.50 | 52.91 | 27.74 | 64.92 | 52.25 | 62.74 | **51.51** |

## E.5 ABLATION STUDY ON COMPRESSION RATE

We analyze the performance of TFCA-Attention across varying compression levels to understand its behavior under different threshold $\tau$ settings. As shown in Table 11, TFCA-Attention maintains strong performance across the LongBench-E benchmark with varying compression ratios. The most aggressive setting ($\tau = 0.3$, 84.69% compression) yields a total reduction of only 1.34% in average performance. Tasks primarily relying on sparse context (Single Document QA, Summarization, and Few-Shot Learning) demonstrate remarkable robustness even at extreme compression rates, while tasks requiring dense global context show mild degradation at the highest compression levels. This demonstrates that TFCA-Attention successfully identifies and preserves the critical information paths needed for these tasks. Tasks requiring dense global context (Multi-Document QA, Synthetic tasks, and Code) show moderate performance degradation at extreme compression. This occurs because these tasks rely on information distributed throughout the entire sequence, making them more sensitive to aggressive token compression.

This differential behavior provides valuable insights into when and how aggressively TFCA-Attention can be applied. For applications where global context integration is critical (e.g., multi-document reasoning), a more conservative compression setting ($\tau = 0.6$ or $0.9$) may be preferable. However, for tasks where only specific passages contain relevant information (e.g., single-document question answering), the highest compression settings can be used with minimal performance impact, maximizing computational efficiency.

Table 11: Performance on LongBench-E at different compression ratios ($\tau$ values).

| Comp. Ratio (%) ($\tau$) | S. QA | M. QA | Sum. | F.S. | Syn. | Code | Avg. |
|--------------------------|-------|-------|------|------|------|------|------|
| 52.72 ($\tau = 0.9$) | 48.50 | 52.91 | 27.74 | 64.92 | 52.25 | 62.74 | 51.51 |
| 73.06 ($\tau = 0.6$) | 48.14 | 49.78 | 27.67 | 64.77 | 51.00 | 62.10 | 50.58 |
| 84.69 ($\tau = 0.3$) | 48.11(-0.39) | 49.34(-3.57) | 27.66(-0.08) | 65.20(+0.28) | 50.00(-2.25) | 60.72(-2.02) | 50.17(-1.34) |

## E.6 ABLATION STUDY ON CALIBRATION DATASET

We conduct an ablation study to evaluate TFCA-Attention's sensitivity to calibration dataset choice and size. As shown in Table 12, our method achieves consistent performance across different calibration datasets and sizes. When calibrated on diverse domains—general web text (SlimPajama), long-form governmental documents (GovReport), and programming code (McEval)—the average score on LongBench-E remains remarkably stable (51.54-51.56). More importantly, this stability persists even with minimal calibration: using just a single input sequence (size=1), our method achieves virtually identical performance to using five samples, with variations of $\leq 0.04$. These results demonstrate two critical properties: (1) head-specific redundancy patterns transfer well across domains, and (2) these patterns are remarkably stable across different inputs within the same model, requiring only minimal calibration data. This efficiency makes TFCA-Attention highly practical for real-world deployment, as it eliminates the need for domain-specific calibration or extensive calibration datasets while maintaining consistent performance across diverse benchmarks (LongBench-E, MMLU, GSM8K, HumanEval).

## E.7 ABLATION STUDY ON BALANCING PARAMETER $\alpha$

We conduct an ablation study to evaluate the sensitivity of TFCA-Attention to the $\alpha$ parameter, which balances the global and local context contributions in our redundancy metric. As shown in Table 13, the performance remains remarkably stable across a wide range of $\alpha$ values (0.1-0.9),

Table 12: Results of Qwen2.5-7B-Instruct on LongBench-E with different calibration datasets and sizes.

| Calibration Dataset | Domain | Average Score |
|---|---|---|
| SlimPajama | General web text | $51.54_{\pm 0.03}$ |
| GovReport | Long-form governmental docs | $51.56_{\pm 0.04}$ |
| McEval | Programming code | $51.55_{\pm 0.02}$ |

demonstrating the robustness of our method to this hyperparameter. The highest average score of 51.51 on LongBench-E is achieved at $\alpha = 0.5$, indicating an optimal balance between global attention patterns and local context dynamics. Notably, even at extreme values ($\alpha = 0.1$ or $\alpha = 0.9$), the performance degradation is minimal ($\leq 0.38$), confirming that TFCA-Attention maintains strong performance without requiring precise parameter tuning. This stability makes our method practical for real-world deployment where automatic parameter selection is preferred over manual optimization.

Table 13: Performance on LongBench-E using Qwen2.5-7B-Instruct with different $\alpha$ values.

| alpha | 0.1 | 0.3 | 0.5 | 0.7 | 0.9 |
|---|---|---|---|---|---|
| Avg. | 51.14 | 51.25 | 51.51 | 51.24 | 51.13 |

### E.8 ABLATION STUDY ON CONCENTRATION INDEX IN THE REDUNDANCY METRIC

We conduct an ablation study to evaluate the effectiveness of the Herfindahl-Hirschman Index in Eq.6. As shown in Table 14, Replacing our Herfindahl-Hirschman Index in Eq.6 with a block-wise entropy measure results in a noticeable performance drop. The Herfindahl-Hirschman Index consistently delivers higher or comparable performance across most tasks, leading to a higher overall average score. This empirically validates that it is a more effective metric for identifying and preserving information-critical context.

Table 14: Performance on LongBench-E using Qwen2.5-7B-Instruct with different Concentration Indexes.

| Methods | S. QA | M. QA | Sum. | F.S. | Syn. | Code | Avg. |
|---|---|---|---|---|---|---|---|
| Entropy | 48.75 | 51.39 | 27.51 | 64.98 | 51.25 | 62.42 | 51.05 |
| Herfindahl-Hirschman Index | 48.50 | 52.91 | 27.74 | 64.92 | 52.25 | 62.74 | **51.51** |

### E.9 COMPUTATIONAL EFFICIENCY ANALYSIS

To understand the computational overhead of TFCA-Attention, we conducted a detailed latency breakdown of our Triton implementation at 128K context length. As shown in Table 15, our block-wise design effectively minimizes the overhead of dynamic token selection components. This efficiency stems from our fundamental design choice to operate at the block level rather than token level. The importance score calculation (8.13%) and other auxiliary operations collectively contribute less than 40% of total latency, demonstrating that our method's adaptive mechanisms introduce only marginal overhead.

## F   LLM USAGE STATEMENT

In preparing this work, the authors used large language models (LLMs) solely to improve the readability and language quality of the manuscript. Specifically, LLMs were employed to assist with:

• Polishing sentence structures and grammatical correctness

• Enhancing the fluency of certain paragraphs

• Ensuring consistent academic tone throughout the paper

Table 15: Latency breakdown of TFCA-Attention components at 128K context length.

| Phase | Improtance Score | Redundancy Metric | Sorting | Token Selection | Attention | Total |
|---|---|---|---|---|---|---|
| Latency (ms) | 32.45 | 46.62 | 42.37 | 39.62 | 238.03 | 399.09 |
| Ratio (%) | 8.13% | 11.68% | 10.62% | 9.93% | 59.64% | 100.00% |

The core research contributions, including the conceptualization of TFCA-Attention, methodological design, theoretical analysis, experimental setup, implementation, and all empirical evaluations, remain entirely our own without any involvement of LLMs.

