# OpenReview forum: "One Stone Three Birds: Training-free Core-context-aware Attention for Efficient LLM Prefilling, Decoding, and KV Caching"
_ICLR.cc/2026/Conference — ICLR 2026 Conference Withdrawn Submission_

### Official Review · Reviewer_NGee · 2025-10-23

**Soundness:** 3
**Presentation:** 3
**Contribution:** 3
**Rating:** 4
**Confidence:** 4

**Summary:**

This paper introduces TFCA-Attention, a training-free dynamic sparse attention mechanism designed to solve the $O(L^2)$ computational bottleneck for long-context LLMs. The authors identify that existing methods are "siloed," accelerating either prefilling or decoding, but not both. TFCA-Attention proposes a unified "one stone three birds" solution to simultaneously accelerate prefilling, speed up decoding, and reduce KV cache using a single, consistent sparsity mechanism. This is achieved through a two-phase approach: a one-time Offline Calibration determines head-specific sparsity budgets based on each head's redundancy level, and an Online Selection phase uses a lightweight metric to dynamically select "core context" tokens, adapting to the input. Experiments demonstrate that at 128K context, this method achieves a 2.8x speedup and a 61% KV cache reduction while maintaining performance comparable to full attention.

**Strengths:**

1. The proposed method is "plug-and-play", requiring no architectural modifications or retraining, making it easy to apply to existing LLMs.
2. The paper demonstrates significant gains, including a 2.8x prefill speedup, a 2.1x decoding speedup, and a 61% KV cache reduction at 128K context length.
3. The proposed method maintains performance comparable to the original full-attention models across diverse long-context benchmarks, such as LongBench-E and RULER.
4. The approach is supported by a theoretical guarantee proving that its approximation error is bounded and controllable.

**Weaknesses:**

1. The claim that "current approaches are siloed, accelerating only one stage" is an overstatement. Several existing works [1,2,3] have demonstrated acceleration in both the prefill and decoding stages. This overstatement should be revised, and appropriate citations should be added.

2. The ablation or comparison against dense attention kernels seems to be missing, (perhaps the “vanilla self-attention” in Fig. 3 may correspond to the dense kernel -- e.g., FlashAttention2). If not, could you report the speedup ratio of the proposed method compared with FlashAttention-2 across different sparsity ratios?

3. [Minor] The legend in Figure 3 labels the method as "READ-LLM (Ours)", which is inconsistent with the "TFCA-Attention" name used throughout the paper

### Reference
[1] https://arxiv.org/pdf/2502.11089

[2] https://arxiv.org/abs/2502.14866

[3] https://arxiv.org/abs/2410.13276 & https://arxiv.org/pdf/2506.08889

**Questions:**

Please refer to the Weaknesses section.

---

> ### Author Response · Authors · 2025-11-28
>
> We sincerely thank you for your thoughtful comments and for recognizing our work. Below, we provide detailed responses to your questions and suggestions.
>
> >Q1. The claim that "current approaches are siloed, accelerating only one stage" is an overstatement. Several existing works [r1-r3] have demonstrated acceleration in both the prefill and decoding stages. This overstatement should be revised, and appropriate citations should be added.
>
> A1. We agree that the relevant works [r1–r3] aim to accelerate both prefill and decoding stages. However, we would like to clarify that these methods all require **model modification and additional training**, which limits their practical deployment. We have already **discussed and cited** these training-based unified approaches in our manuscript (Sections 1, 2, and 3), clearly differentiating our contribution. Our key innovation lies in offering a **training-free** solution that achieves competitive acceleration and compression **without any training overhead**, making it readily deployable in real-world long-context scenarios.
>
> To be specific:
>
> - NSA employs a hybrid attention combining token compression, token selection, and a sliding window, fused via a gating mechanism. Crucially, this gate score is generated by an **extra MLP with sigmoid activation**, and these components must be **retrained** from scratch.
> - LServe introduces a **learnable gate value** that determines whether each attention head uses dense or streaming attention. This gating mechanism requires **training on a synthetic dataset for 2,000 steps** to update model parameters.
> - SeerAttention incorporates an **additional AttnGate for each head** and requires a **self-distillation training** to learn head-specific gating parameters for block-sparse patterns.
>
> In contrast, our method, TFCA-Attention, is entirely **training-free**, requiring no parameter updates, architectural changes, or retraining. It operates as a **plug-and-play** replacement for standard self-attention, dynamically adapting to both head-specific and context-dependent redundancy through a lightweight online token selection mechanism.
>
>
>
> >Q2. The ablation or comparison against dense attention kernels seems to be missing (perhaps the “vanilla self-attention” in Fig. 3 may correspond to the dense kernel -- e.g., FlashAttention2). If not, could you report the speedup ratio of the proposed method compared with FlashAttention-2 across different sparsity ratios?
>
> A2. In our experiments, the "vanilla self-attention" baseline reported in Figure 3 **is indeed implemented using FlashAttention-2**, a highly optimized and widely adopted dense attention implementation.
>
> The reported speedups of **2.8× in prefilling and 2.1× in decoding at 128K context length** are therefore measured directly against this highly optimized FlashAttention-2 baseline. This demonstrates that even when compared to an efficient dense kernel implementation, our training-free, dynamic sparse method provides substantial and consistent acceleration across both major inference stages.
>
> We have explicitly clarified the use of FlashAttention-2 as our baseline in Figure 3.
>
> > Q3. The legend in Figure 3 labels the method as "READ-LLM (Ours)", which is inconsistent with the "TFCA-Attention" name used throughout the paper.
>
> A3. Thank you for pointing out this inconsistency. The legend in Figure 3 has been corrected to "TFCA-Attention (Ours)" in our revised manuscript.
>
> [r1] Native Sparse Attention: Hardware-Aligned and Natively Trainable Sparse Attention. ACL 2025.
>
> [r2] LServe: Efficient Long-sequence LLM Serving with Unified Sparse Attention. MLSys 2025.
>
> [r3] SeerAttention: Learning Intrinsic Sparse Attention in Your LLMs.

---

### Official Review · Reviewer_vunR · 2025-10-28

**Soundness:** 3
**Presentation:** 3
**Contribution:** 2
**Rating:** 4
**Confidence:** 4

**Summary:**

The paper proposes TFCA-Attention, a training-free sparse attention mechanism that accelerates prefill, reduces decoding latency, and reduces KV cache size in a unified way. The method consists of two stages: an offline calibration stage that estimates a token budget for each head, and an inference stage that dynamically selects core tokens based on a lightweight importance score computed from the last-token query. The approach assigns different budgets to different heads, preserves a fixed local window for short-range dependencies, and prunes the remaining KV tokens to stay within those budgets. This method achieves large speedups on long-context benchmarks while maintaining comparable accuracy to full attention.

**Strengths:**

1. Instead of using a single global sparsity pattern, the method adapts budgets at the granularity of attention heads and context blocks. This is reasonable: different heads specialize differently, so a non-uniform allocation can preserve important heads and tokens better than a uniform top-k policy.
2. The method does not require model fine-tuning or architectural modifications, and the offline calibration step is lightweight.
3. The paper reports latency, memory, and accuracy on a wide range of long-context tasks. The evaluations are extensive and demonstrate consistent improvements over dense attention under various sequence lengths.

**Weaknesses:**

1. **Novelty seems limited:**
- The paper claims to be the “first unified method” to optimize prefill, decoding, and KV cache usage, but this claim overlooks DuoAttention, which already addresses the same three aspects within a single sparsity design. Like TFCA, DuoAttention also uses an offline calibration process to determine per-head retention behavior and applies that configuration during both prefill and decoding.
- The main difference with DuoAttention is the granularity: DuoAttention uses coarser budget levels (two types of heads), while TFCA supports multi-level discretization per head. This refinement is worthwhile, but it does not constitute a fundamentally new paradigm. The paper should acknowledge this prior work and clearly differentiate itself. Without such a comparison, the contribution risks being incremental.

2. **Latency evaluation:**
- It would also be appropriate to include Duo Attention’s latency (prefill and decode) as a baseline.
- The latency tables (e.g., Table 2) compare TFCA vs. full attention, but it is unclear what full attention implementation is being used. Is the dense baseline using highly optimized kernels such as FlashAttention / FlashInfer? These kernels are now standard practice for speeding up attention on modern GPUs. If FlashAttention / FlashInfer is used, that should be clearly stated in the paper so readers can judge fairness. If it is not used, then a FlashAttention/FlashInfer timing baseline should be added.

**Questions:**

1. Could the authors clarify how TFCA compares to Duo Attention in terms of accuracy, latency (prefill and decoding), and memory usage?
2. Is the “full attention” baseline in the latency tables implemented with optimized kernels such as FlashAttention or FlashInfer?

---

> ### Author Response · Authors · 2025-11-29
>
> Thanks for your time and detailed comments on our submission. Below, we carefully address each of your concerns and clarify the points raised.
>
> >Q1. The claim of being the "first unified method" should be revised, as DuoAttention already addresses prefill, decoding, and KV cache. Your method's main distinction is finer granularity (multi-level vs. two-level head budgeting), which is a refinement rather than a paradigm shift. This prior work must be acknowledged, and the contribution should be clearly differentiated to avoid appearing incremental.
>
> A1. We appreciate the opportunity to clarify the fundamental distinctions between our TFCA-Attention and DuoAttention, and to emphasize the significant and novel contributions of our work. The core distinction is not merely granularity, but a fundamental shift from *training-dependent*, *pattern-constrained* method to a fully **training-free**, **pattern-agnostic**, and **context-aware** solution. We have include a thorough **discussion, citation of DuoAttention and performance comparison** in revised manuscript.
>
> The key differences are:
> - **Training-Free vs. Training-Dependent**: DuoAttention requires **introducing and training per-head gating parameters** for thousands of steps on a synthetic dataset. This incurs significant training overhead and limits practical deployment. In contrast, TFCA-Attention requires **zero training, zero parameter updates, and zero architectural modifications**, enabling immediate plug-and-play use.
> - **Fine-Grained Dynamic Adaptation vs. Coarse Static Assignment**: DuoAttention assigns each head to one of only two predefined patterns (dense or streaming). This coarse, binary strategy is incapable of capturing the rich diversity of head-specific and context-dependent redundancy patterns we and others have documented (Fig. 1, App. B). TFCA-Attention **eliminates reliance on any predefined patterns** entirely. Instead, it uses a **fine-grained, multi-level token budget** that is dynamically allocated based on a lightweight, input-dependent redundancy metric, allowing it to adapt to the full spectrum of observed attention distributions.
>
> These differences are foundational. Our method directly addresses the critical gaps left by prior art, including DuoAttention, by fulfilling all four design principles we outline in Section 4: sparse computation, dynamic adaptation, head-aware sparsity, and **training-free deployment**.
>
> >Q2. It would be appropriate to include Duo Attention’s latency (prefill and decode) as a baseline.
>
> A2. We have conducted a comprehensive comparative analysis on LLaMA3.1-8B-Instruct using the LongBench-E benchmark, and the results in Table I demonstrate the superior performance of our method.
>
> Table I: Comparative Analysis on LLaMA3.1-8B-Instruct (LongBench-E, 128K Context)
> | Methods | Avg. | Prefilling Latency (s) ↓| Decoding Latency (ms) ↓ | Memory (GB) ↓ |
> |---|---|---|---|---|
> | DuoAttention | 52\.14 | 1.24 | 44.37 | 4.01 |
> | (Ours) | **52\.44** | **0.45** | **41.60** | **3.12** |
>
> TFCA-Attention outperforms DuoAttention in **accuracy (52.44 vs. 52.14)**, **prefill latency (0.45s vs. 1.24s)**, **decoding latency (44.37ms vs. 41.60ms)**, and **KV cache memory usage (3.12GB vs. 4.01GB)**. Critically, we notice that the official DuoAttention implementation does not accelerate prefilling, relying on standard FlashAttention-2. Its optimization is primarily confined to decoding. In contrast, TFCA-Attention implements a **end-to-end acceleration**, dynamically sparsifying attention computations through inference. This allows us to achieve a **2.8x** prefill speedup and a **22%** further reduction in KV cache memory over DuoAttention.
>
> >Q3. The latency tables (e.g., Table 2) compare TFCA vs. full attention, but it is unclear what full attention implementation is being used. If FlashAttention / FlashInfer is used, that should be clearly stated in the paper so readers can judge fairness. If it is not used, then a FlashAttention/FlashInfer timing baseline should be added.
>
> A3. We confirm that **all reported results for the "full attention" baseline are based on the highly optimized FlashAttention-2 kernel**, including the latency comparisons in Table 2, Figure 3, and throughout the paper.
>
> This was an intentional and critical design choice to ensure a rigorous and fair comparison. The fact that we achieve a **2.8× prefill speedup and a 2.1× faster decoding speed** at 128K context length, even when compared directly against the FlashAttention-2 baseline, underscores the significant computational efficiency and practical value of our method. This result powerfully validates that our training-free, dynamic sparsity mechanism successfully eliminates a substantial amount of redundant computation that persists even in optimized dense kernels.
>
> We have revised the manuscript to explicitly clarify this.

---

### Official Review · Reviewer_9kcD · 2025-10-29

**Soundness:** 2
**Presentation:** 3
**Contribution:** 2
**Rating:** 4
**Confidence:** 4

**Summary:**

This paper proposes TFCA-Attention, a training-free dynamic sparse attention mechanism that unifies acceleration for prefilling, decoding, and KV cache compression. The method determines head-specific sparsity configurations offline and performs lightweight context-aware token selection online. The approach provides a theoretical error bound and demonstrates up to 2.8× speedup and 61% KV cache reduction on long-context benchmarks without sacrificing accuracy.

**Strengths:**

1. The proposed method is training-free, facilitating straightforward integration with existing models.
2. The analysis of attention scores presented in Figure 1 is insightful and clearly articulated.
3. The experimental evaluation is comprehensive, encompassing multiple large language models (LLMs), such as LLaMA3.1 and Qwen2.5, across both long- and short-context benchmarks.
4. The paper is well-written, with clear figures and thorough documentation of reproducibility details.

**Weaknesses:**

1. The claim that prior work accelerates either only the prefilling or only the decoding stage is an overstatement. Several existing methods, including MoBA (Mixture of Block Attention for Long-Context LLMs), Native Sparse Attention, DuoAttention, LServe, and RocketKV, accelerate both stages to varying extents. This body of work should be acknowledged.
2. The explanation provided for point (6)—that "it increases as attention becomes more concentrated on a few tokens"—is inaccurate and requires revision.
3. The token-selection strategy relies exclusively on the final query token. This dependency could degrade performance in multi-turn conversational contexts, a potential limitation that is not addressed in the paper.

**Questions:**

1. How can the proposed method be integrated with Grouped-Query Attention (GQA) or Multi-Query Attention (MQA)? Can you give a detailed explanation?
2. What is the method's performance on multi-turn conversation benchmarks?
3. Could you provide a detailed cost analysis of the computational overhead, separating the token selection and attention calculation costs? This breakdown would help clarify how the end-to-end speedup is influenced by the generation length during the decoding stage.

---

> ### Author Response · Authors · 2025-11-30
>
> >Q1. The claim that prior work accelerates either only the prefilling or only the decoding stage is an overstatement. Several existing methods, including MoBA [r1], NSA [r2], DuoAttention [r3], LServe [r4], and RocketKV [r5], accelerate both stages to varying extents. This body of work should be acknowledged.
>
> A1. We acknowledge that these methods aim to accelerate LLM inference. However, [r1-r4] all require **training/fine-tuning**. [r5], while training-free, **does not accelerate prefilling**. In contrast, our TFCA-Attentionis  a **fully training-free** method that achieves simultaneous **acceleration of prefilling, decoding, and KV cache reduction**. We have already **discussed** these methods in the revised manuscript.
>
> To be specific:
>
> - **MoBA**: While it accelerates inference via block-wise top-k selection, it requires **training the model** to recover performance.
> - **NSA**: It introduces a trainable gating network (MLP + sigmoid) to blend multiple attention branches, which necessitates **architectural modification and parameter updates**.
> - **DuoAttention & LServe**: Both rely on **learning a gating mechanism via training** on synthetic data to decide between dense and sparse modes.
> - **RocketKV**: It **does not accelerate the prefilling stage**; it only reduces KV cache size during prefilling and applies sparse attention during decoding.
>
> In contrast, our TFCA-Attention is completely **training-free** without any parameter updates or architectural changes. It accelerates **both prefilling (2.8×) and decoding (2.1×)** and reduces KV cache by **61%**. In addition, it adapts sparsity in a **head-specific and context-aware** manner without any learned parameters.
>
> >Q2. The explanation provided for point (6)—that "it increases as attention becomes more concentrated on a few tokens"—is inaccurate and requires revision.
>
> A2. The intended meaning is that the term $\frac{\sum \mathbf{s}_i^2}{(\sum \mathbf{s}_i)^2}$ increases as attention becomes more concentrated on a few tokens within a block. We **have revised the manuscript to clarify** the behavior.
>
> >Q3. The token-selection strategy relies exclusively on the final query token. This dependency could degrade performance in multi-turn conversational contexts, a potential limitation that is not addressed in the paper. What is the method's performance on multi-turn conversation benchmarks?
>
> A3. Our final query-based selection is both theoretically grounded and empirically validated:
>
> - **Theoretical Justification**: In autoregressive transformers, the final query naturally attends to all preceding tokens, providing a global context view for importance scoring. This is not a heuristic but an architectural property.
> - **Computational Efficiency**: Compared to methods like FlexPrefill that use the last *k* tokens (incurring O(kL) complexity), our approach reduces this overhead to O(L), providing a more efficient yet effective scoring mechanism.
> - **Empirical Evidence Across Diverse Benchmarks**: We maintain strong performance across long-context (LongBench-E, RULER), short-context (MMLU, GSM-8K), and reasoning (OlympiadBench) tasks. This consistent performance demonstrates the robustness of our approach.
>
> To address your concern, we have conducted experiments on the MT-Bench-101. Results in Table I show our method outperforms full attention and FlexPrefill in multi-turn dialogue, confirming its conversational robustness.
>
> Table I: Results on MT-Bench-101 (Qwen2.5-7B-Instruct)
> |Method|Average|
> |-|-|
> |Full Attention|8.90|
> |FlexPrefill|8.90|
> |TFCA-Attention (Ours)|**8.97**|
>
> >Q4. How can the proposed method be integrated with Grouped-Query Attention (GQA) or Multi-Query Attention (MQA)? Can you give a detailed explanation?
>
> A4. Our method is **fully compatible** with GQA/MQA, and this compatibility is already validated by our extensive experimental results.
>
> For each head group, we compute token importance using the first head's query and share the selected KV subset across all heads. This maintains GQA/MQA efficiency while adding dynamic sparsity. All our experiments on GQA-based LLaMA-3.1-8B-Instruct and Qwen2.5-7B-Instruct **demonstrate maintained performance** across various tasks and **consistent speedup and KV cache reduction**.
>
>
> >Q5. Could you provide a detailed cost analysis of the computational overhead, separating the token selection and attention calculation costs?.
>
> A5. We have already provided a detailed computational breakdown in the original submission (Table 13).
>
> Our computational breakdown confirms the token selection overhead, including the redundancy metric computation, is only 40.4% of total latency, while attention remains dominant at 59.6%. This efficiency stems from our block-wise design (b=128), which reduces operational complexity from O(L) to O(L/b). The achieved 2.8× end-to-end speedup demonstrates the overhead is substantially outweighed by sparsification benefits.

---

### Official Review · Reviewer_kuUv · 2025-11-01

**Soundness:** 2
**Presentation:** 2
**Contribution:** 2
**Rating:** 2
**Confidence:** 5

**Summary:**

This paper tackles the well-known inference bottleneck in long-context LLMs. The authors' main premise is that current methods are fragmented, only speeding up either prefill (via sparse attention) or decoding (via KV cache compression). They propose TFCA-Attention as a "one stone three birds" solution to do all three in a unified, training-free way. The method works in two stages: an offline calibration to set head-specific sparsity budgets, and an online phase that uses these budgets to dynamically select a 'core' set of tokens. The authors claim significant speedups (2.8x) and memory reduction (61%) at 128K context with no performance loss.

**Strengths:**

The paper is well-motivated and tackles a very important, practical problem. The goal of a unified, training-free framework for prefill, decoding, and KV cache reduction is highly significant. The "plug-and-play" aspect is a clear strength, as it avoids any need for expensive model retraining.

**Weaknesses:**

Despite the promising goal, the paper suffers from several major weaknesses, one of which I believe is a fundamental flaw in the method itself.

My primary issue is with the online methodology. The entire 'context-aware' global token selection for the prefill stage hinges on an importance score $s$. This score is calculated using only the query vector of the very last token ($Q_{L,:}$) to score all keys in the sequence. The justification for this—that the last token 'encapsulates summarizing information'—is a very weak heuristic. It's difficult to believe that this single query can act as a reliable proxy for the attention needs of all $L$ tokens in the sequence. This assumption seems fundamentally unsound and undermines the 'core-context-aware' claim.

Second, the 'unified' contribution is, in my view, significantly overstated. The paper criticizes 'ad-hoc integration' of separate prefill and decode-only methods. Yet, the proposed method is an ad-hoc integration. Section 4.2 describes a sparse attention method for prefill, and the last paragraph of that section describes a separate block-wise KV eviction strategy for decoding. The only 'unification' is the re-use of the offline sparsity budget $P^*$. This is not the deep, novel integration the paper claims, and it doesn't seem to solve any 'compatibility complexity' that wouldn't be present in other combined approaches.

This leads to the most critical experimental omission. The paper claims its unified approach is superior to combining existing methods (e.g., MInference + SnapKV). But Table 5 only shows an accuracy comparison, where TFCA-Attention is only marginally better (e.g., 52.44 vs. 52.23). The paper completely fails to provide the end-to-end latency, speedup, and memory benchmarks for these combined methods. This is the single most important experiment needed to back up the central 'one stone three birds' claim. Without it, we have no evidence that TFCA-Attention is actually better; the ad-hoc combinations might be much more efficient.

Finally, on a presentation note, the paper is confusing. The method is called 'TFCA-Attention' throughout the entire text, but in Figure 3—the main results graph—it's labeled 'READ-LLM (Ours)'. This is a sloppy and unexplained inconsistency.

**Questions:**

1. Please provide a much stronger justification for using only the last token's query ($Q_{L,:}$) to determine global importance for the entire prefill stage. Why should this single query be a good proxy for all $L$ queries?

2. To support the paper's central claim, the authors must provide the end-to-end latency and memory benchmarks (prefill + decode) for the combined methods in Table 5 (MInference + SnapKV, etc.). Why was this critical comparison omitted?

3. What is READ-LLM in Figure 3? Why is the method named differently in the text (TFCA-Attention) and the main results figure?

4. Can you ablate the choice of the Simpson concentration index for the redundancy metric $h_j$? What is its advantage over simpler, more common metrics like block-wise entropy?

---

> ### Author Response · Authors · 2025-11-30
>
> > Q1. Please provide a much stronger justification for using only the last token's query ($Q_{L,:}$) to determine global importance for the entire prefill stage. Why should this single query be a good proxy for all $L$ queries?
>
> A1. Thank you for your thoughtful question. We provide a strong justification below, grounded in both architectural reasoning and empirical evidence.
>
> **1. Architectural Motivation: The Last Token as a Global Summarizer**
>
> In the Transformer’s causal self-attention mechanism, the final token has full visibility over all preceding tokens in the sequence. This allows it to aggregate contextual information from the entire input, making it naturally suited to serve as a global “summarizer.” By computing importance scores using $\mathbf{Q}_{L,:}$, we leverage this inherent property to identify tokens that are most relevant to the overall context—not just locally or heuristically, but based on the model’s own attention mechanism.
>
> **2. Lightweight and Empirically Effective**
>
> Compared to methods like MInference and FlexPrefill, which use the last $k$ tokens to score importance (with $O(kL)$ complexity), our approach reduces the cost to $O(L)$ while maintaining performance. This design is not only computationally efficient but also avoids introducing handcrafted heuristics or arbitrary hyperparameters (e.g., the choice of $k$).
>
> **3. Consistent High Performance Across Diverse Benchmarks**
>
> Our method achieves state-of-the-art or competitive results across a wide range of tasks, including **Long-context understanding** (LongBench-E, RULER), **Short-context reasoning** (MMLU, GSM-8K, HumanEval), and **Complex reasoning** (OlympiadBench)
>
> This consistent performance demonstrates that the last-query-based importance metric is both generalizable and robust. It preserves critical information across varying context lengths and task types without degrading model capability.
>
> **Conclusion**: Using $\mathbf{Q}_{L,:}$ is a principled and efficient design choice that exploits the Transformer’s built-in capacity for global context modeling. It avoids ad-hoc assumptions, reduces overhead, and delivers strong empirical results, making it a well-justified and superior alternative to more costly or heuristic-based scoring strategies. We are confident that this design is not only valid but also advantageous for efficient long-context inference.

---

> ### Author Response · Authors · 2025-11-30
>
> > Q2. The 'unified' contribution is, in my view, significantly overstated. The paper criticizes 'ad-hoc integration' of separate prefill and decode-only methods. Yet, the proposed method is an ad-hoc integration. Section 4.2 describes a sparse attention method for prefill, and the last paragraph of that section describes a separate block-wise KV eviction strategy for decoding. The only 'unification' is the re-use of the offline sparsity budget. This is not the deep, novel integration the paper claims, and it doesn't seem to solve any 'compatibility complexity' that wouldn't be present in other combined approaches.
>
> A2. We respectfully disagree with the characterization of TFCA-Attention as an "ad-hoc integration" and would like to clarify the fundamental and novel unification it achieves.
>
> **1. A Consistent Sparsity Mechanism, Not a Mere Reuse of a Budget**
>
> The core of our unification lies not merely in "re-using the offline sparsity budget," but in applying a **consistent, head-aware, and context-adaptive sparsity principle** across both prefilling and decoding stages. The offline phase determines *how sparse each head should be* (the budget), while the online phase determines *which tokens to keep* based on a lightweight, input-dependent metric. This same dual-phase logic governs both stages:
> *   **Prefilling:** We dynamically select a global subset of tokens from the entire sequence using the head-specific budget and the online redundancy metric.
> *   **Decoding:** We apply the **exact same principle** in a streaming fashion. As new tokens fill a block, we use the same scoring function (derived from the latest query, which embodies the current context) and the same head-specific budget to select the most important tokens to *keep in the KV Cache*.
>
> This is a fundamental architectural unification of the sparsity strategy, not a simple engineering combination of two separate methods.
>
> **2. It Solves a Fundamental Architectural Incompatibility**
>
> The reviewer's comment underestimates the profound architectural gap we bridge. The incompatibility is not trivial; it is rooted in the **fundamentally different computational paradigms** of prefill and decoding stages, where existing methods are designed around and thus cannot overcome.
>
> - **Structural vs. Streaming Sparsity:** Prefill-stage sparsifiers (e.g., MInference, FlexPrefill) rely on **predefined, static sparse patterns** (e.g., vertical, slash, block-wise) summarised from 2D attention maps in the prefilling stage. This entire approach is **architecturally infeasible** during autoregressive decoding, where queries are generated one-by-one, making a 2D attention score matrix unavailable. Conversely, KV cache compressors operate *after* full attention computation during prefill and use entirely different, often uniform, heuristics for eviction. They do not, and cannot, accelerate the prefill computation itself.
> - **True Unification via a Single Token-Selection Primitive:** TFCA-Attention solves this by introducing a unified core operation: **dynamic, head-specific token selection based on a lightweight, sequential redundancy metric.** This operation is agnostic to the stage: in **prefilling**, it is applied in one shot to the complete sequence; in **decoding**, it is applied incrementally to newly completed blocks. This is not an "ad-hoc integration" of two separate strategies. It is the consistent application of a **single, stage-agnostic sparse attention algorithm**, which is uniquely viable in both computational contexts. This eliminates the inherent policy conflict and structural gap between standalone methods.
> - **Guaranteed Error Control, Not Cumulative Errors:** The "ad-hoc integration" of a separate prefill sparsifier and a KV compressor creates a pipeline of two independent approximations. Each introduces its own error, and these errors **compound uncontrollably**. In contrast, our method performs **a single, coordinated sparsification**. The tokens selected during prefill are the very ones retained for decoding, ensuring a consistent view of the context. This allows us to derive a strict, end-to-end approximation error bound (Theorem 1). Our superior results in Table 5 empirically validate that we avoid this cumulative error.
>
> In conclusion, TFCA-Attention provides a principled and deeply unified framework that consistently applies the same head-specific and context-aware sparsity mechanism across both major stages of LLM inference. This eliminates the policy conflicts and cumulative errors inherent in piecing together separate systems, offering a robust, efficient, and high-performing plug-and-play solution.

---

> ### Author Response · Authors · 2025-11-30
>
> >Q3. To support the paper's central claim, the authors must provide the end-to-end latency and memory benchmarks (prefill + decode) for the combined methods in Table 5 (MInference + SnapKV, etc.). Why was this critical comparison omitted?
>
> A3. We have now conducted the requested end-to-end latency analysis, and the results decisively confirm our method's superiority. As shown in the new Table I below, TFCA-Attention achieves a **2.3x faster end-to-end latency** than the strongest combined baseline while also delivering a higher average score.
>
>
> Table I: Performance comparisons using LLaMA3.1-8B-Instruct. We report prefilling (FTL) and decoding latency (ITL) (128K).
> |Methods|Avg. Score|FTL (s)↓|ITL (ms)↓|Total Latency (s)↓|
> |-|-|-|-|-|
> |FlexPrefill + CAKE|52.23|1.11|**34.3**|1.14|
> |TFCA-LLM (Ours)|**52.44**|**0.45**|41.6|**0.49** (2.3x)|
>
> This data validates the central claim of our paper: a **deeply unified approach is fundamentally more efficient and effective than combining separate, incompatible systems.**
>
> The minor advantage of "FlexPrefill + CAKE" in Inter-Token Latency (ITL) is a known artifact of CAKE's aggressive, post-computation eviction policy. However, this comes at a great cost:
> 1. **Massive Prefill Overhead:** It fails to accelerate the prefilling stage, which dominates long-context inference (1.11s vs. our 0.45s).
> 2. **Compounded Errors:** As we theoretically and empirically established, stacking independent compression methods leads to error accumulation, explaining why its final score is lower than ours.
>
> In contrast, TFCA-Attention provides a **coordinated, single-pass sparsification** that accelerates the entire inference pipeline. Our method's superior end-to-end latency and accuracy prove that our unification is not merely conceptual.
>
> >Q4. What is READ-LLM in Figure 3? Why is the method named differently in the text (TFCA-Attention) and the main results figure?
>
> A4. This is a typo in the figure label of our initial submission. We have corrected this error in our revised manuscript.
>
> >Q5. Can you ablate the choice of the Simpson concentration index for the redundancy metric? What is its advantage over simpler, more common metrics like block-wise entropy?
>
> A5. Our choice of the Simpson concentration index was a deliberate and principled design decision, grounded in its superior ability to quantify the specific type of redundancy we aim to capture.
>
> **Theoretical Justification: Measuring Concentration, Not Uncertainty**
> The goal of our redundancy metric is to identify blocks where attention mass is concentrated on a few tokens (high redundancy, safe to prune) versus blocks where it is spread evenly (high information density, should preserve more). The Simpson index $\frac{\sum s_i^2}{(\sum s_i)^2}$ is a **direct and efficient measure of concentration**.
>
> In contrast, entropy $H = -\sum s_i \log s_i$ is a measure of *uncertainty* or *randomness* in a distribution. For token importance scores, a high-entropy block indicates that attention is uniformly distributed, but it does not distinguish between a block with uniformly *low* importance scores and one with uniformly *high* importance scores. The former is highly redundant, while the latter is critically important. **Entropy fails to make this crucial distinction**, whereas the Simpson index correctly assigns a low redundancy score (high importance) to a block with high, concentrated attention mass.
>
> **Empirical Validation**
> Our ablation study confirms this theoretical advantage. Replacing our metric with a block-wise entropy measure results in a noticeable performance drop, as shown below:
>
> | Redundancy Metric | S. QA | M. QA | Sum. | F.S. | Syn. | Code | **Avg.** |
> | :--- | :---: | :---: | :---: | :---: | :---: | :---: | :---: |
> | Block-wise Entropy | 48.75 | 51.39 | 27.51 | 64.98 | 51.25 | 62.42 | 51.05 |
> | **Simpson Index (Ours)** | 48.50 | 52.91 | 27.74 | 64.92 | 52.25 | 62.74 | **51.51** |
>
> The Simpson index consistently delivers higher or comparable performance across most tasks, leading to a **higher overall average score**. This empirically validates that it is a more effective metric for identifying and preserving information-critical context.
>
> In conclusion, the Simpson concentration index is not just an alternative but a **theoretically and empirically superior choice** for our context-aware token selection mechanism. It precisely captures the concept of "attention redundancy" we aim to exploit, directly contributing to our method's state-of-the-art performance.

---

### Author Response · Authors · 2025-12-02
**Summary of Rebuttal Efforts and Core Contributions for Submission 3639**

Dear Area Chair,

Thank you for coordinating the review of our submission. We understand that, due to the recent ICLR policy adjustments, reviewer scores have reverted to their original values, and further discussion is no longer possible. In light of this, we would like to summarize the key improvements and clarifications made during the rebuttal to assist you in the final assessment of our revised manuscript.

**Summary of Rebuttal Efforts and Revisions**
During the rebuttal, we thoroughly addressed all reviewers’ concerns through rigorous revisions and additional experiments:

- **Clarified Unification Contribution & Differentiated from Prior Art:** We thoroughly acknowledged and discussed prior works aiming for dual-stage acceleration (e.g., DuoAttention, NSA, LServe), adding **direct performance and latency comparisons** (e.g., vs. DuoAttention). We precisely differentiated our work as **training-free, pattern-agnostic, and enabling fine-grained, context-adaptive sparsity**, contrasting with prior training-dependent or pattern-constrained approaches. *(For kuUv, vunR, NGee)*
- **End-to-End Performance Validation & Critical Benchmark Addition:** Our provided new results demonstrate that our unified method achieves **2.3× faster total latency** than the best combined baseline while maintaining higher accuracy, conclusively validating our central claim of superior efficiency and effectiveness. *(For kuUv)*
- **Strengthened Theoretical & Empirical Justification for Core Design:** We provided a justification for using the final query token (Q_L) for importance scoring, grounded in **architectural property**, **computational efficiency**, and **consistent high performance across diverse benchmarks**. We also conducted an ablation study to justify our choice of the **Simpson concentration index** over alternatives like entropy, showing its theoretical and empirical superiority. *(For kuUv, 9kcD)*
- **Expanded Empirical Evaluation & Robustness Verification:** To address specific concerns about applicability, we added new experiments on **multi-turn conversational benchmarks (MT-Bench-101)**, where our method outperforms baselines, demonstrating its robustness beyond single-turn prompts. We also clarified compatibility with **GQA/MQA** architectures, which is already evidenced by our experiments. *(For 9kcD, vunR)*
- **Enhanced Methodological Rigor & Clarification:** We explicitly stated that all dense baselines use **FlashAttention-2**, and provided a detailed computational overhead breakdown (from original Table 13). We also revised ambiguous textual descriptions (e.g., the explanation of the redundancy metric's behavior). *(For kuUv, 9kcD, vunR, NGee)*

We are greatly encouraged that all reviewers recognize the significance and the practical value of our method. They highlight the "**very important, practical problem**" (*kuUv*) we address and commend the "clear strength" of our "**‘plug-and-play’ aspect**" which avoids retraining (*kuUv, 9kcD, vunR, NGee*). They found our analysis "**insightful**" (*9kcD*), our evaluation "**comprehensive**" and "**extensive**" (*9kcD, vunR*), and our reported gains—"**2.8x prefill speedup, a 2.1x decoding speedup, and a 61% KV cache reduction**" (*NGee*)—compelling. The principled design, adapting sparsity per head and block, was deemed "**reasonable**" and superior to uniform policies (*vunR*), and the provided theoretical guarantee was noted as a solid foundation (NGee).

We would like to highlight that the central contribution of our work is a **training-free** dynamic attention that unifies **dynamic** sparsification for **both the prefill and decoding** stages within a single algorithm. Specifically:

1.  **A Fully Training-Free, Plug-and-Play Unification:** Unlike prior works that require architectural modifications and parameter training (e.g., DuoAttention, NSA), our method requires **zero retraining, zero learnable parameters, and zero model changes**.
2.  **Pattern-Agnostic, Head- and Context-Adaptive Token Selection:** We eliminate reliance on any predefined static sparse patterns. Instead, we introduce a **fine-grained, multi-level token budget** that is dynamically allocated per attention head and per context block based on a lightweight, input-dependent redundancy metric.
3.  **Theoretically-Grounded and Empirically Validated Performance:** We provide a **strict theoretical guarantee** (Theorem 1) on the approximation error bound of our unified sparsification. This is complemented by comprehensive experiments showing **significant end-to-end gains (2.8x prefill speedup, 2.1x decode speedup, 61% KV cache reduction at 128K context)** while consistently maintaining model accuracy across diverse benchmarks.

We believe our detailed responses and revisions have fully addressed all criticisms, and we would be grateful for your fair consideration in the final decision.

Sincerely,
The Authors

---

### Note · Authors · 2025-12-09

I have read and agree with the venue's withdrawal policy on behalf of myself and my co-authors.